# Investigating future changes in the volume budget of the Arctic sea ice in a coupled climate model

Ann Keen[1] and Ed Blockley[1]

[1]Met Office, FitzRoy Road, Exeter, EX1 3PB, United Kingdom

*Correspondence to*: Ann Keen (ann.keen@metoffice.gov.uk)

**Abstract.** We present a method for analysing changes in the modelled volume budget of the Arctic sea ice as the ice declines during the 21$^{st}$ century. We apply the method to the CMIP5 global coupled model HadGEM2-ES to evaluate how the budget components evolve under a range of different forcing scenarios. As the climate warms and the ice cover declines, the sea ice processes that change the most in HadGEM2-ES are summer melting at the top surface of the ice due to increased net

downward radiation, and basal melting due to extra heat from the warming ocean. There is also extra basal ice formation due to the thinning ice. However, the impact of these changes on the volume budget is affected by the declining ice cover. For example, as the autumn ice cover declines the volume of ice formed by basal growth declines as there is a reduced area over which this ice growth can occur. As a result, the biggest contribution to Arctic ice decline in HadGEM2-ES is the reduction in the total amount of basal ice growth during the autumn and early winter.

Changes in the volume budget during the 21$^{st}$ century have a distinctive seasonal cycle, with processes contributing to ice decline occurring in May/June and September to November. During July and August the total amount of sea ice melt decreases, again due to the reducing ice cover.

The choice of forcing scenario affects the rate of ice decline and the timing and magnitude of changes in the volume budget components. For the HadGEM2-ES model and for the range of scenarios considered for CMIP5, the mean changes in the

volume budget depend strongly on the evolving ice area, and are independent of the speed at which the ice cover declines.

## 1 Introduction

Arctic September sea ice cover has declined at a rate of over 13% per decade since satellite observations began (Serreze and

Stroeve, 2015), and the ice that remains is becoming thinner (Kwok and Rothrock, 2009), younger (Maslanik et al, 2011), and faster moving (Rampal et al, 2009, Spreen et al, 2011). The ice cover is projected to reduce further as greenhouse gas concentrations continue to increase (Stroeve et al, 2012). These changes have implications both within the Arctic itself, for example for shipping (Melia et al, 2016), and local ecology (Post et al, 2013), and also for the wider climate system via large scale circulation changes that have been linked to the reducing Arctic ice cover (Francis et al 2009, Overland and Wang, 2010).

As the sea ice interacts directly with both the atmosphere and the ocean, it is influenced by changes in both, and as such can be seen as an integrator of wider changes within the Arctic region.

Hence there is much interest in how the decline in Arctic sea ice will continue in the future, both in terms of the predictability of ice cover in a given year, and in terms of the manner and timing of the transition to a seasonally ice-free Arctic. Global coupled models are arguably the best tool we have for making future projections of the Arctic sea ice, but generate a wide spread of projections of future ice decline (Stroeve et al 2012). There are many factors potentially contributing to this spread, including sea ice model formulation, forcing from atmosphere and ocean model components, uncertainty in forcing scenarios, and internal model variability. A number of studies have attempted to decrease the spread of plausible future projections by sub-selecting models based on their ability to simulate current day sea ice (Wang and Overland, 2009), or past observed changes (Massonnet et al, 2012). More recent work has focussed on the role of internal model variability (Jahn et al, 2016), and the extent to which it is realistic to expect modelled ice decline to closely match the observed decline (Notz, 2015).

Given the inherent uncertainties in predicting future changes in 'integrated' quantities like ice cover and volume, it is becoming increasingly clear that it is also necessary to consider, compare and evaluate the underlying processes causing ice growth and decline, and how they are likely to change in a warming world. Holland et al (2010) evaluated the annual mean changes in ice growth, melt and divergence during the 21$^{st}$ century for a range of models submitted to the CMIP3 model archive, finding considerable variation in the magnitude and relative importance of changes in the budget components. For this 2010 study, the budget components were derived from model monthly ice thickness and velocity from the CMIP3 data archive. However, for individual models a more detailed decomposition is often possible (eg Keen et al, 2013), and for CMIP6 models a wide range of budget components should be available for intercomparison (Notz et al, 2016). In addition, new process-based observational datasets are becoming available to help understand whether the modelled ice state arises for the right reasons (Holland and Kimura, 2016, Uotila et al, 2014).

In this study we introduce a method for analysing how the modelled volume budget of the Arctic sea ice (and overlying snow) changes during the 21$^{st}$ century. The data required for this decomposition forms part of the data request for the CMIP6 Sea-Ice Model Intercomparison Project (SIMIP), and so for the next generation of climate models this method can be used for model inter-comparison (see Notz et al, 2016, and https://www.earthsystemcog.org/projects/wip/CMIP6DataRequest). Here we use a CMIP5 model for which the budget components are already available as model output, and consider the processes contributing to 21$^{st}$ century changes in the volume of the Arctic sea ice and overlying snow in the Met Office Hadley Centre model HadGEM2-ES (Martin et al. 2011; Collins et al. 2011). We use a similar budget formulation to Holland et al (2010), so that components of the volume budget are expressed in terms of their impact on the mean ice thickness over a defined domain of the Arctic. The data available to Holland et al (2010) only allowed a decomposition between advective, melt and freeze processes, and only considered the annual mean changes. Here we are able to decompose the budget further into individual processes causing ice growth and loss, and we also consider the seasonal cycle of the volume budget. The application of the method allows us to investigate how the volume budget evolves during the 21$^{st}$ century, and to identify the dominant processes contributing to the decline in ice volume. We also evaluate how the declining ice area impacts the changes in the volume

budget, and consider how key budget changes relate to wider changes in the Arctic and beyond. As HadGEM2-ES projections are available for a range of different 21$^{st}$ century forcing scenarios, we also evaluate the impact of forcing scenario on the evolving volume budget.

In summary, the scope of this work is to introduce our method of analysing the volume budget of the Arctic sea ice, and to use the method to learn about 21$^{st}$ century changes in the HadGEM2-ES model. In Sect. 2 we describe the model and the forcing scenarios used. In Sect. 3 we describe the mean volume budget for this model, and in Sect. 4 we investigate how this changes during the 21$^{st}$ century for a range of forcing scenarios. In Sect. 5 we summarise and discuss our findings.

## 2 Model description and integrations used

### 2.1 Model description

HadGEM2-ES is a coupled atmosphere-ocean model that was submitted to CMIP5. The model includes interactive atmosphere and ocean carbon cycles, dynamic vegetation, and tropospheric chemistry (Martin et al. 2011; Collins et al. 2011). It is considered to have a good depiction of present-day global cloud characteristics (Jiang et al, 2012) and the best model depiction of Arctic cloud and surface radiative forcing (English et al 2015). The mean Arctic ice extent lies within 20% of observed values at all time of year, although September extent is biased low and the magnitude of the seasonal cycle is too large, consistent with biases in winter net surface LW and summer net surface SW (West et al, 2018)

The horizontal resolution of the atmosphere component is 1.25° latitude by 1.875° longitude, with 38 vertical levels. The ocean component is 1° by 1° outside the tropics, increasing to 0.33° latitude by 1° longitude at the equator, and has 40 vertical levels. The sea ice formulation within HadGEM2-ES is essentially the same as the one used in HadGEM1 (McLaren et al., 2006), with three updates as follows:

- The bare sea ice albedo was increased from 0.57 to 0.61, together with a correction to sea-ice albedo during surface melt.

- Heat fluxes passed from the atmosphere to the ocean/seaice model are regridded taking the ice concentration into consideration.

- Sea ice velocities are combined with ocean currents to create a "surface velocity" field for use in the atmosphere model.

Some of the sea ice calculations take place within the atmosphere component, where the sea ice surface temperature and the top melting and diffusive heat fluxes are computed using the zero-layer thermodynamics scheme described by Semtner (1976). In this scheme the sea ice has no heat capacity, and the ice and any overlying snow are treated as one layer with an effective thickness $h_e$ defined as

$$h_e = h + (\kappa_i / \kappa_s) h_{s,} \tag{1}$$

where $h$ is the ice thickness, $\kappa_i$ and $\kappa_s$ are the (constant) thermal conductivities of ice and snow, and $h_s$ is the snow depth. The

albedo of the sea ice is a function of surface temperature (Curry et al, 2001), allowing the radiative impact of melt ponds to be represented in a simple way.

The growth and melt of ice is calculated within the ocean component, and the ocean to ice heat flux is calculated following McPhee (1992). There is a sub-gridscale ice thickness distribution (Thorndike et al, 1975), with 5 thickness categories plus open water, and the thermodynamic transfer of ice between categories is calculated using a linear remapping scheme (Lipscomb, 2001). Ice velocities are calculated following the elastic-viscous-plastic (EVP) model of Hunke and Dukowicz (1997), using the Hibler (1979) formulation for ice strength. The amount of ridging is determined following the approach used in the CICE model (Lipscomb and Hunke, 2004). For a fuller description of the HadGEM1 sea ice component, see McLaren et al (2006).

## 2.2 Model integrations

The integrations used here are described in Jones et al (2011), and include an ensemble of 4 historical simulations (Hist) using observed forcing from 1860 to 2005, and initialised from the model state at 50 year intervals of a pre-industrial control integration. Four different climate forcing scenarios developed for the IPCC Fifth Assessment Report (AR5) (Moss et al, 2010) were then run from the end of each of these historical simulations, providing an ensemble of four simulations for each forcing scenario. Here we consider the period 1960 to 2099 (comprising part of the historical period, followed by the scenario). Fig. 1 shows the global temperature anomalies for these HadGEM2-ES integrations w.r.t. a reference period taken as the years 1960-89. There is little divergence in the global temperature response before the middle of the 21$^{st}$ century, but by 2100 the temperature increase relative to 1960-89 ranges from less than 2 degrees for RCP2.6 to nearly 5.5 degrees for RCP8.5.

## 2.3 Evolution of ice area and volume

We focus on changes in the sea ice over the domain shown in Fig. 2, covering the Arctic basin and the Barents Sea. Figure 3 shows how the ice area and volume within this domain declines for each of the model integrations during the period 1960 to 2090. The ice volume is expressed as a mean thickness over the domain, calculated as the total ice volume within the domain divided by the area of the domain. The impact of any overlying snow is included by converting the snow to an equivalent thickness of ice using the ratio of ice and snow conductivities from Eq. (1). This is added to the ice to create an *effective ice thickness*. Hereafter, whenever ice thickness or volume is mentioned it refers to this *effective* value, which includes the overlying snow as well.

The March ice area over the domain declines from a mean value of 9.3 x10$^6$ km$^2$ during the 1960-89 reference period, to 8.4 x10$^6$ km$^2$ towards the end of the 21$^{st}$ century (2090-2099) for the RCP2.6 scenario, and 5.2 x10$^6$ km$^2$ for the more aggressive RCP8.5 scenario (Fig. 3a). There is little divergence in the response of either the ice area or volume to the different forcing scenarios before about 2050 (Fig. 3), after which the stronger forcing scenarios show a greater loss of winter ice cover, with RCP8.5 showing an especially steep decline from 2080 onwards. This rapid decline in winter ice cover is seen in other climate models as well (Bathiany et al, 2016). It occurs once the summer ice in the Arctic Ocean has gone, and when regions of the

central Arctic Ocean no longer fall to the freezing temperature over the winter. The seasonal ice can no longer form at these locations, leading to a rapid drop in the winter ice cover.

The mean March ice thickness over the domain declines from 2.3m during the period 1960 to 1989, to 1.2m during the 2090s for the RCP2.6 scenario, and 0.2m for RCP8.5.

For September, the mean ice area during the 1960-89 reference period is 4.0 x10$^6$ km$^2$, and the mean thickness is 1.0m. By the end of the 21$^{st}$ century, all the scenarios have less than 1.0 x10$^6$ km$^2$ of ice cover remaining in September, so that the Arctic Basin is virtually ice free.

## 3 Mean volume budget of the Arctic sea ice

The HadGEM2-ES model output includes sea ice volume tendencies due to thermodynamic and dynamic processes, and terms
quantifying the thermodynamic processes acting on the ice and overlying snow. This allows us to construct a budget that balances the diagnosed changes in ice volume over any given period. In Keen et al (2013), the budget terms are expressed in terms of a heat anomaly per unit area of ice (in J m$^{-2}$). While this formulation enables an understanding of how the atmospheric and oceanic forcing of the ice is changing as the climate warms, the budget terms expressed this way cannot be summed to balance the changes in the ice volume. Here we start by expressing the budget components in terms of their impact on the
average ice thickness over the domain of Fig. 2, so the units are m of ice formed/lost. This is a similar formulation to that used by Holland et al (2010).

### 3.1 Mean volume budget for the reference period 1960-89

The components of the volume budget that we can diagnose for the HadGEM2-ES model are shown in Fig. 4, both as a decadal mean time series (for the RCP8.5 scenario) and as a mean seasonal cycle for the reference period 1960-89. As mentioned
above, each component is expressed in terms of its impact on the ice thickness (averaged over the domain): a flux representing heat entering the ice will be shown as a negative value as it causes ice loss. We describe each component in turn:

- Basal ice growth via the diffusive heat flux through the ice and snow (dark green lines) : Ice growth is dominated by basal ice formation due to the loss of heat via the diffusive heat flux through the ice and snow (Fig. 4a). This term is positive for most of the year (Fig. 4b), representing ice growth at the base of existing ice. The total amount of basal
growth increases as ice forms during the autumn, and is a maximum during the winter, reaching 29cm of ice growth during December. During the summer, this term can become small and negative (representing ice melt) when the surface temperature rises about the freezing temperature of sea water (Fig. 4b).

- Basal melting due to heat from the ocean (light green lines): The ocean to ice heat flux is a function of the difference in temperature between the top layer of the ocean, and the temperature at the base of the ice (McPhee, 1992). It is

maintained through diffusive and advective ocean processes, and melts ice at the bottom surface throughout the year, especially during the summer and autumn (Fig. 4b). This term is small and negative during the winter, increasing in magnitude from April to a maximum of 24cm of ice loss in July, and then declining in magnitude through the late summer and autumn. This is the largest individual term causing ice melt (Fig. 4a).

• Top melting (dark blue lines): The top melting flux is the sum of the atmospheric turbulent and radiative heat fluxes, resulting in the surface melting of ice or snow. It is zero outside the melting season (Fig. 4b), and negative during the spring and early summer (as it causes ice melt). The amount of top melting peaks in June at 37cm of ice loss. The maximum occurs earlier in the melt season than the basal melting, and then declines more quickly. There is less ice lost during the year by top melting than by basal melting (Fig. 4a).

• Advection (orange lines): The net impact of ice advection is to move ice out of the domain (to lower latitudes), and so this appears as a negative term in Fig. 4. There is a small seasonal cycle, with more ice lost by advection during the winter, falling from a monthly maximum of 2.7cm of ice loss during January, to 0.8cm by August. The amount of ice lost by advection each decade is smaller than the amount lost by either top or basal melting (Fig. 4a).

     • Frazil ice formation (green/blue lines): This term represents the formation of ice in open water when the ocean
temperature would otherwise fall below the freezing temperature. This new ice is formed with a specified local thickness of 0.05m, and is added to the first ice thickness category. After this 0.05m ice has formed, any additional freezing would be counted as basal ice growth. Frazil ice formation is virtually zero during the summer, and a maximum in autumn (2.5cm of ice formation during November) as the ocean cools and the ice cover increases following its summer minimum. This component is always positive, as it solely represents ice formation.

• Snowfall (less sublimation) (red lines): This represents the snow accumulation due to snowfall, less any loss of ice or snow at the surface due to sublimation. It is positive in all months, a maximum during the winter (1.2cm of ice formation in December), and virtually zero during the summer melt season.

     To summarise, in the decadal mean volume budget for HadGEM2-ES, ice growth is dominated by basal ice formation due
to the diffusive heat flux through the snow and ice, which accounts for 85% of the annual mean ice formation during the reference period 1960-89. There are smaller contributions due to frazil ice growth (7%) and the accumulation of snow (less sublimation) (7%). These processes are offset by melting at the base of the ice due to heat from the ocean (48% of annual mean ice loss), melting at the top of the ice due to atmospheric fluxes (40%), and ice advection out of the region (12%).
     The sum of the budget components (black line, Fig. 4a) is much smaller in magnitude than the individual components,
representing the near balance between the processes of ice growth and loss. The ice decline arises because of the small

imbalance between these terms in the warming climate. By magnifying the budget sum in Fig. 4a (black dash line), we can see that after the first few decades the sum is always negative, representing the ongoing loss of ice. This budget sum matches the decadal changes in ice thickness seen in Fig. 3b.

The mean volume budget for HadGEM2-ES described here is very similar to the budget for our previous model HadGEM1, shown in Keen et al. (2013). This is perhaps not surprising as the two models have near-identical sea ice physics. There are some subtle differences however: there is more surface melting during June in HadGEM2-ES, consistent with the positive net SW bias at this time of year (West et al; 2018). In addition, HadGEM1 has thicker and more extensive Arctic ice than HadGEM2-ES (Martin et al, 2012), leading to less basal melting in the late summer due to reduced ocean heating, and less
winter ice growth due to the thicker ice.

The HadGEM2-ES melting season extends from May to September during the 1960-89 reference period (Fig. 4b, solid black line). During this time the melting is initially dominated by melting at the top surface of the ice, with basal melting due to heat from the ocean becoming more important later in the melt season, and continuing into the autumn. During the winter, the dominant term is basal ice growth due to the diffusive heat flux.

Note that ridging is not included in this decomposition, as it does not explicitly affect the ice volume: it changes the spatial distribution of ice within a grid box, but not the volume of the ice. That is not to say that the ridging is unimportant, merely that it has a null direct impact on the volume budget. In addition, lateral melting is not explicitly modelled and so does not appear in this decomposition, although for low ice concentrations there is an adjustment to the ocean to ice heat flux to provide a crude representation of lateral ice melt of small ice floes. Finally, as we are considering the combined budget of the ice and
overlying snow there is no snow-ice formation term.

To summarise, in this section we have defined and quantified the mean volume budget for HadGEM2-ES during the reference period 1960-89, and identified the most important processes. Next, we will examine how this budget changes during the subsequent decades as greenhouse gas concentrations increase.

**4 Changes in the volume budget of the Arctic sea ice**

Here we consider how the components in the volume budget change relative to the reference period 1960-89 discussed in Sect. 3, both in terms of their decadal evolution during the 21[st] century, and the changes in the seasonal cycle. Initially we focus on the strongest forcing scenario RCP8.5, and then we consider the impact of the different forcing scenarios on the changes.

**4.1 Budget changes for the RCP8.5 forcing scenario**

Figure 5 shows how the components of the volume budget change relative to the reference period 1960-89 for the RCP8.5 scenario. As the ice starts to decline, the ice loss initially results from a mean reduction in basal ice formation due to the

diffusive heat flux through the ice (dark green line, Fig. 5a), and extra melting at the base of the ice due to heat from the ocean (light green line). There is also a reduction in the accumulation of falling snow on the ice (red line). These changes are shown as negative values in Fig. 5, representing less ice growth (or more ice loss) relative to 1960-89. Offsetting these are a reduction in melting at the top surface of the ice due to atmospheric fluxes (dark blue line), reduced loss by advection (orange line), and

more frazil ice formation (green/blue line).

As the run progresses, the majority of these changes become more pronounced as the ice cover declines, the exceptions being the basal melting and the frazil ice formation (Fig. 5b), which each change sign. The amount of frazil ice formation initially increases (Fig. 5a), then after 2010 it begins to decrease, until by the 2050s there is less frazil ice formation than during 1960-89 (Fig. 5b). The total amount of basal melting initially increases relative to 1960-89 (Fig. 5a), and then decreases from 2010

onwards, until by the 2040s there is less basal melt than there was during 1960-89 (Fig. 5b). In each case the reversal in sign is due to alterations in the balance between opposing changes that occur at different times of year, and is most easily understood by looking at changes in the seasonal cycles of the budget components (Fig. 6)

The budget changes causing extra net ice loss relative to 1960-89 occur at two distinct times of year: during May/June, and again during September-November (black line, Fig. 6a). These are partially offset by the changes occurring at other times of

year, most notably during July and August.

Early in the melt season (May/June) there is extra ice loss due to top and basal ice melt, and also due to reduced basal ice growth (Fig. 6a). In contrast, during July and August there is less top melting and no extra basal melting, and so less net ice melt relative to the reference period. During the autumn, there is reduced basal ice formation, which becomes the largest budget change resulting in ice loss.

The changes shown in Fig. 6a are for the decade 2010-2019. Later in the integration, changes in the budget components show broadly the same seasonal pattern as for the earlier decade, and the magnitude of the changes relative to 1960-89 increases as the ice area declines (Figs. 6b and 6c). During the 2040s (Fig. 6b), the most notable differences are that the amount of basal melt in the late summer has reduced relative to the reference period, and there is now no net change in the amount of ice loss during June. Then towards the end of the 21$^{st}$ century (Fig. 6c), the reduction in the amount of basal ice growth extends into

the winter months, and during June there is a reduction in the volume of ice lost by surface melting.

The amount of ice lost by advection is reduced at all times of year, and to a greater extent during the winter (orange line, Fig. 6). This is consistent with the reducing ice volume – there is less ice that can move out of the basin. In fact by the 2060s (not shown), there is virtually no advective ice loss during August and September, consistent with the Arctic basin being almost completely free of ice by the end of the summer (Fig. 3).

There is reduced frazil ice formation in the autumn during the 2040s (Fig. 6b), and an increase in the winter months (November to March). The autumn change is consistent with warmer temperatures delaying the freeze-up, and the winter change is consistent with decreased ice cover exposing a larger area of ocean where frazil ice can form. During the following decades, as the ocean surface continues to warm, the reduction in frazil ice formation continues later into the winter months (Fig. 6c).

## 4.2 Impact of the declining ice area on the volume budget

Having described how the volume budget evolves during the 21$^{st}$ century, we now consider the processes contributing to these changes. As the climate warms, the processes causing ice formation and loss will change accordingly. As the ice cover declines, the impact of these process changes on the volume budget will be modified by changes in the ice area. For example, suppose partway through the 21$^{st}$ century the amount of basal melting per unit area of the ice during September has doubled compared to the reference period. If the September ice cover has reduced by half over the same period then the volume budget will show no net change in the total amount of basal melt that month. As the ice cover declines more quickly during the late summer and autumn than at other times of year (Figs. 7a and 7b), we might expect the evolving ice cover at this time of year to have the greatest impact on the volume budget. In Fig. 7c, the dominant budget components are weighted by the ice area to give a *local* value showing the change per unit area of ice during the decade 2010-19. Using this decade as an example, we consider each of these processes in turn to see how it changes as the climate warms, and how the declining ice area affects its impact on the volume budget.

Top melting

During May and June there is more local melting at the surface of the remaining ice during 2010-19 than there was during the reference period 1960-89. In contrast, at other times of year there is little change in the local surface melting (Fig. 7c). The changes in top melting are primarily driven by changes in the surface SW and LW fluxes. Over the entire Arctic region considered here, approximately 74% of the mean increase in the net downward radiative flux at the ice and ocean surface during May is due to changes in SW radiation. The incoming SW decreases as the atmosphere warms, due to increased water vapour and changes in the impact of cloud. The outgoing SW also decreases, partly because there is less incoming SW, but predominantly because of the reduced surface albedo (67%). Hence the outgoing SW decreases by a greater amount than the incoming SW, and so the net effect of the SW changes is extra surface melting. In common with other CMIP5 models, incoming LW increases due to the higher levels of $CO_2$ in the atmosphere (Notz and Stroeve (2016) found a robust linear relationship between incoming longwave fluxes and cumulative $CO_2$ emissions for CMIP5 models). The extra incoming LW is almost solely due to the extra atmospheric $CO_2$: there is little change in the impact of cloud on downward LW. Outgoing LW also increases as the surface temperature warms, but as this increase is smaller than the increase in incoming LW the balance results in more net downward LW, leading to extra surface melting.

The impact of the local top melting changes on the volume budget is modified by the associated changes in the ice area (Fig. 6a). In the 2010s, during May and June the extra local melt over the remaining ice dominates, leading to a net loss of ice in the volume budget. However, during July and August there is no extra local melting over the remaining ice, and as the ice area has reduced this means a smaller volume of ice melts compared to the reference period. This appears as a net gain of ice in the volume budget.

As the model integration continues, the declining ice area has more and more impact on the top melting component of the volume budget. By the 2040s there is a smaller volume of ice melted during June as well as during July and August.

Basal melting

As the Arctic Ocean warms, there is more local melting at the base of the ice throughout the melt season, especially during July, August and September (Fig. 7c). Year on year, the warming of the Arctic Ocean in HadGEM2-ES is driven by ocean heat transport from lower latitudes, with a net heat loss due to atmospheric surface fluxes (Burghard and Notz, 2017). However,

during spring and summer, a budget analysis of the upper ocean shows that atmospheric fluxes cause a strong *warming* of the ocean surface, and this is the dominant process warming the upper ocean during the melt season. Then during the autumn and winter, the ocean loses more heat to the atmosphere than it gained over the spring and summer, resulting in the net annual heat loss due to atmospheric fluxes. This ocean budget analysis follows the approach used by Graham and Vellinga (2012), who found a similar result for previous Met Office models HadCM3 and HadGEM1.

Therefore, the extra local basal melting seen in Fig. 7c is primarily due to the in-situ warming of the ocean surface as the ice cover retreats. Comparing Fig. 7c and Fig. 6a, we see that in the 2010s the extra local melting at the base of the remaining ice during May and June translates into a greater total amount of ice loss in the volume budget. In contrast, during July-September the volume budget for 2010-19 shows no extra ice loss due to basal melt. This is because of the larger reduction in ice area compared to 1960-89 at this time of year (Fig. 7b). As a result, the extra local basal melting over the remaining ice in 2010-19

cannot compensate for the impact of the reduced ice cover, and so the volume budget shows no extra ice loss during July-September. As the model integration continues and the ice cover declines further, its effect on this term in the budget becomes more dominant. By the 2020s the volume budget has *less* basal ice melt compared to the reference period during August (illustrated in Fig. 6b), and by the 2050s this is the case for July and September as well (illustrated in Fig. 6c).

These contrasting seasonal changes explain the evolution of the decadal changes shown in Fig. 5. Until the 2020s, the impact

of the extra local basal melting over the remaining ice dominates, and the decadal budget shows a net ice loss due to basal melt (w.r.t. the reference period). Later in the integration, the impact of the declining ice area dominates, and the decadal budget shows a net ice *gain* due to changes in basal melt.

Basal ice growth

From October through to March, there is extra local ice growth at the base of the remaining ice during the 2010s w.r.t. the

reference period (Fig. 7c). In contrast, between May and September, there is reduced local ice growth compared to the reference period. The diffusive heat flux causing the basal growth is a function of the surface temperature and the ice thickness: colder surface temperatures or thinner ice result in more ice growth. At lower surface temperatures there is a stronger dependence on the ice thickness, so that from October onwards there is more diffusive heat loss to the atmosphere, and more local basal ice formation. However, between May and September the impact of the warmer atmosphere and ice surface dominates, resulting

in a smaller diffusive heat flux and less local ice growth.

Again, the impact of these process changes on the volume budget depends on the declining ice area. Within the volume budget, the largest changes due to basal growth occur during September, October and November (Fig. 6a). During September, the reduced ice area in the 2010s amplifies the impact of the reduced local basal ice growth on the volume budget. In contrast, during October and November, although there is *more* local basal ice growth, the impact of the declining ice area dominates

so that the total volume of ice grown is reduced compared to the reference period to give a net ice loss. By the 2080s, the sharp decline in winter ice cover seen in Fig. 3a results in the net ice loss due to changes in basal growth extending into the winter (Fig. 6c).

In summary, for the HadGEM2-ES model there is increased local melting at the top and bottom surfaces of the Arctic sea ice in the spring and summer respectively as the climate warms in response to the RCP8.5 forcing scenario (Fig. 7c). As the ice thins, there is increased local basal ice growth. In the volume budget, the impact of these local changes is affected by the declining ice area. Until the 2020s the ice volume budget shows ice loss due to reduced amounts of basal ice growth and extra basal melting. These decreases in ice volume are offset by a reduced volume of ice being melted at the top surface, and reduced advective ice loss. However, later in the 21st century the total amount of basal melting decreases due to the shrinking ice area in the late summer, so that in the volume budget the basal melt term changes sign to represent a net gain in ice volume relative to the reference period (Fig. 6b).

## 4.3 Impact of forcing scenario

We now consider how the volume budget changes for the other forcing scenarios. Figure 8 shows changes in basal growth (relative to the reference period) for each of the four scenarios. All the scenarios show a decline in basal ice growth, with the more aggressive scenarios showing a greater decline. For the latter half of the 21st century there is a clear difference in response between RCP2.6/4.5 and RCP6.0/8.5. For RCP2.6 and RCP4.5, the amount of basal ice growth levels off towards the end of the 21st century, consistent with the stabilisation of the ice area and volume in these scenarios (Fig. 3). In contrast, for RCP6.0 the amount of basal growth continues to decline throughout the 21st century, albeit to a lesser extent than for the stronger RCP8.5 scenario (Fig. 8). The steep decline in the amount of basal ice growth in RCP8.5 during the latter part of the 21st century is due to the sharply declining winter ice cover at this time. As previously mentioned, areas of the Arctic Ocean become too warm for ice to form during the winter months, thus reducing the area over which basal ice formation can occur. There is also an associated sharp reduction in frazil ice formation at this time (not shown).

A similar picture emerges for the other budget components (not shown): the signals of change for each scenario are broadly the same as already described for RCP8.5, although the exact timing and magnitude of the changes depends on the strength of the forcing.

By plotting the decadal response in each budget component as a function of decadal mean ice area rather than time (Fig. 9), we see that they each follow a common trajectory independent of the forcing scenario. Note that the plots in Fig 9 have different scales, as the intention here is to show the trajectory of each component, rather than their relative magnitudes. Hence, changes in the volume budget components are independent of the speed at which the ice retreats, at least for the HadGEM2-ES model, and for the range of IPCC scenarios considered here. Figure 9 also confirms that the changes in the volume budget components are, as previously discussed, strongly dependent on the ice area.

In contrast to the thermodynamic budget components, the changes in effective ice thickness due to advection are relatively small in relation to the inter-decadal variability of the control integration. By the end of the 21$^{st}$ century, the ratio between the response for scenario RCP8.5 and the variability in the control run is 6.8 for the advective term, whereas for the other budget terms this ratio ranges from 21.7 to 34.9.

We note that for most of the budget components the relationship with the ice area is non-linear. For example, when the annual mean ice area has reduced to approximately 6.5 x10$^6$ km$^2$, the anomaly w.r.t 1960-89 in the total amount of frazil ice formation and basal melting changes sign, and the slope in the response of the basal ice formation steepens. This corresponds to the stage at which the Arctic basin first becomes seasonally ice-free, as shown in Fig. 10 where the budget components from Fig. 9 are plotted against the appropriate (10 year mean) September ice area. As the Arctic becomes seasonally ice-free, processes that

were initially dominant in the ice volume budget during late summer/early autumn have a reduced impact on the decadal mean budget.

Although not shown here, the seasonal cycle of anomalies in the volume budget is also related to the remaining ice cover, and independent of the speed at which the ice retreated. For example, if we choose a decade for each of the scenarios with matching mean ice cover and over plot the anomalies they are very similar.

In summary, while the strength of the forcing scenario affects the magnitude and timing of the modelled decline in ice area during the 21$^{st}$ century, for HadGEM2-ES the changes in the volume budget at any chosen time during the scenario depend on the remaining ice cover, and are independent of the speed at which the ice retreated.

## 5 Summary and discussion

We have presented a method for investigating changes in the volume budget of the Arctic sea ice as the ice declines due to

increasing greenhouse gas forcing. Our approach is distinct from previous work as we are able to include terms representing the individual processes causing ice growth and loss, and we consider the seasonal cycle of changes to show the (sometimes opposing) changes at different times of year. The budget is constructed so that the sum of the budget terms balances the changes in the ice volume. This mean that the declining ice cover has to be taken into account when summing the terms, so that changes in the budget depend both on changes in processes and changes in ice cover. To help distinguish between the two, we also

evaluate how the dominant processes in the budget change locally over the remaining ice cover.

The method has been used to investigate changes in the volume budget of the climate model HadGEM2-ES during the 21$^{st}$ century for a range of IPCC forcing scenarios. For this model, the local processes that change most as the climate warms are top melting and basal melting during the summer and autumn. Extra local top melting occurs during May and June, while local basal melting due to extra heat from the warming ocean occurs from May onwards, reaching a peak in August and September.

When the declining ice area is taken into account, so that the budget terms can be summed to balance the actual changes in ice volume, we see that the decline in ice volume results primarily from a reduction in ice growth, offset by smaller reductions in

ice melt and reduced advection to lower latitudes. Holland and Landrum (2015) have also noted the influence of the declining ice area on processes contributing to 21st century changes in the sea ice.

The seasonal cycle of the ice volume budget shows net ice loss in the spring and early summer due to extra surface and basal melting and reduced basal ice growth, and in autumn/early winter due to reduced basal growth. These changes are partially
offset by net ice gain due to reduced surface melting during July and August as the ice cover declines.

The budget changes shown here are likely to be dependent on the sea ice physics. For example, the fact that the sea ice albedo is a function of surface temperature means that no further albedo reduction are possible once the surface temperature has reached the melting point. A different behaviour may be seen in a model including an explicit representation of melt ponds. Also HadGEM2-ES uses zero-layer thermodynamics, which does not model the internal temperature of the ice, and has a
constant ice salinity. A model including a multi-layer thermodynamic scheme and prognostic salinity might well show a greater sensitivity to the forcing scenario. Finally, HadGEM2-ES does not have an explicit representation of lateral melting and so this term does not appear in our budget. Early results from our CMIP6 model HadGEM3 GC3.1 (Williams et al, 2018) show lateral melting to be an important component of the volume budget during June, July and August, with values of up to 14% of the ocean to ice heat flux.

We have found a strong (non-linear) relationship between the declining ice area and the evolution of the volume budget components, which holds over the range of forcing scenarios considered for IPCC AR5. In common with other climate models (Stroeve and Notz, 2015), for HadGEM2-ES there is a linear relationship between the Arctic ice area and the global near-surface temperature. In addition, the CMIP5 models show a linear relationship between Arctic ice area and cumulative $CO_2$ emissions (Notz and Stroeve, 2016). Hence the relationship found here between the evolving volume budget terms and the ice
area indicates there is also a strong connection with the amount of $CO_2$ emitted, and with the wider climate response to increasing $CO_2$. If this relationship proves to be robust across models, we may in the future be able to derive strong links between emitted $CO_2$ and the processes causing ice decline.

For the next generation of climate models, we will be able to establish the extent to which the changes found here are also seen in other models. The model diagnostics required for this analysis form part of the SIMIP data request for CMIP6 (Notz et al.,
2016), and so the method presented here can be utilised as a model inter-comparison tool for the CMIP6 models. This will mean that we can quantify, for each model, not only how the ice declines, but also why.

Previous work using a more limited range of model output from CMIP3 models found a large inter-model spread in both the present day ice mass budget, and in the magnitude and relative importance of changes in the ice melt and growth terms over the 21[st] century (Holland et al, 2010). For CMIP6 models, we will be able to further decompose the ice volume budget and
establish whether improvements to the representation of sea ice processes have led to a closer agreement in how the volume budget evolves as the climate warms.

*Data availability:* Monthly mean ice and snow thickness, ice area, surface radiation and ocean temperatures for HadGEM2-ES are available from the CMIP5 data archive: https://cmip.llnl.gov/cmip5/data_portal.html.

*Competing interests*. The authors declare no competing interests

*Acknowledgements*. This work was supported by the Joint UK BEIS/Defra Met Office Hadley Centre Climate Programme (GA01 101), and the European Union's Horizon 2020 Research & Innovation programme through grant agreement No. 727862 APPLICATE. The content of the paper is the sole responsibility of the authors and it does not represent the opinion of the European Commission, and the Commission is not responsible for any use that might be made of the information. We thank Tim Graham for making code for calculating the heat budget of the Arctic Ocean available, and Helene Hewitt and Jeff Ridley for helpful discussions during the course of this work. We also thank two anonymous reviewers for their useful comments and suggestions.

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

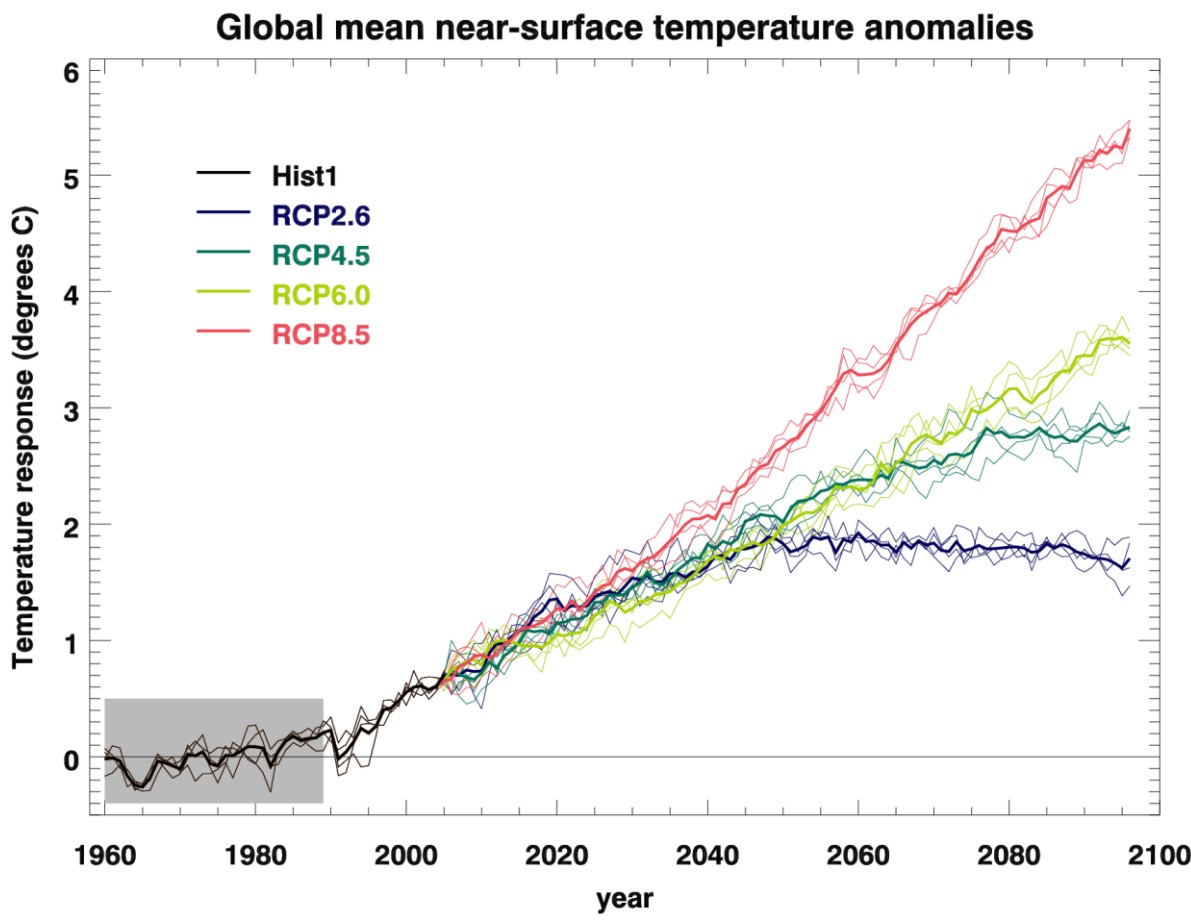

**Figure 1:** Global mean near-surface air temperature anomalies for HadGEM2-ES for the IPCC CMIP5 Historical forcing scenario (black), followed by RCP8.5 (red), RCP6.0 (light green), RCP4.5 (dark green), and RCP2.6 (blue). The shaded region indicates the 1960-89 reference period. Bold lines show the ensemble means, and thin lines show the individual ensemble members in each case.

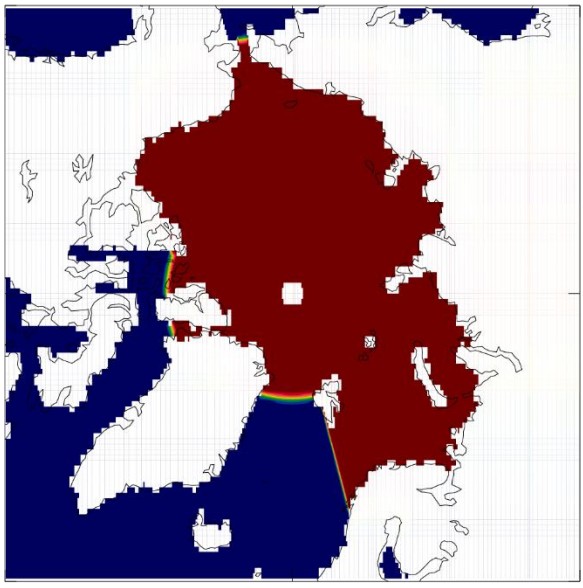

**Figure 2:** The region of the Arctic used in the analysis. For HadGEM2-ES this is formed by masking out all data south of 65N for all latitudes and then the area bounded by 65N to 78N and 90W to 15E.

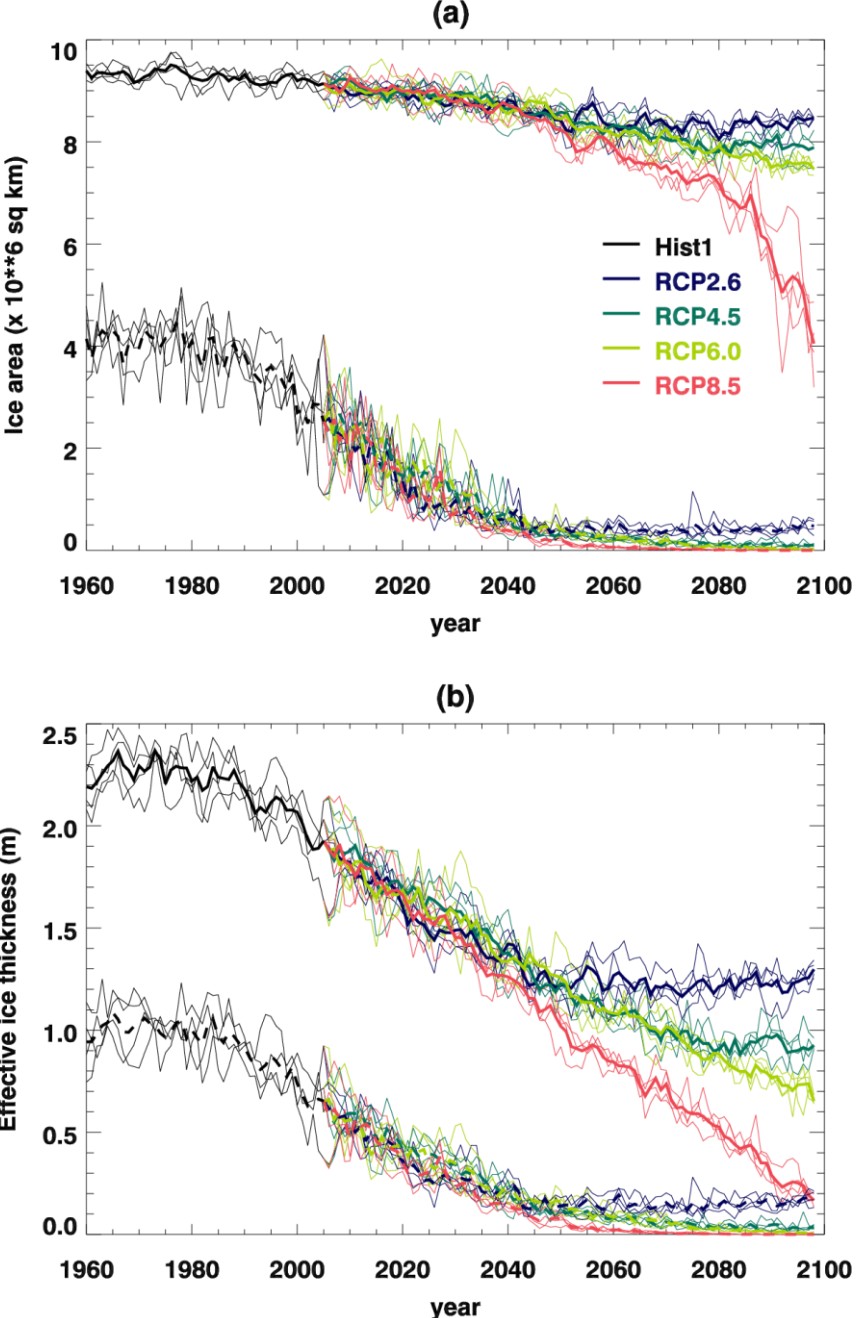

**Figure 3:** Evolution of (a) the ice area and (b) the mean effective ice thickness for March (solid lines) and September (dash lines) over the region defined in Fig. 2, for each of the HadGEM2-ES integrations. Bold lines show the ensemble means, and thin lines show the individual ensemble members in each case.

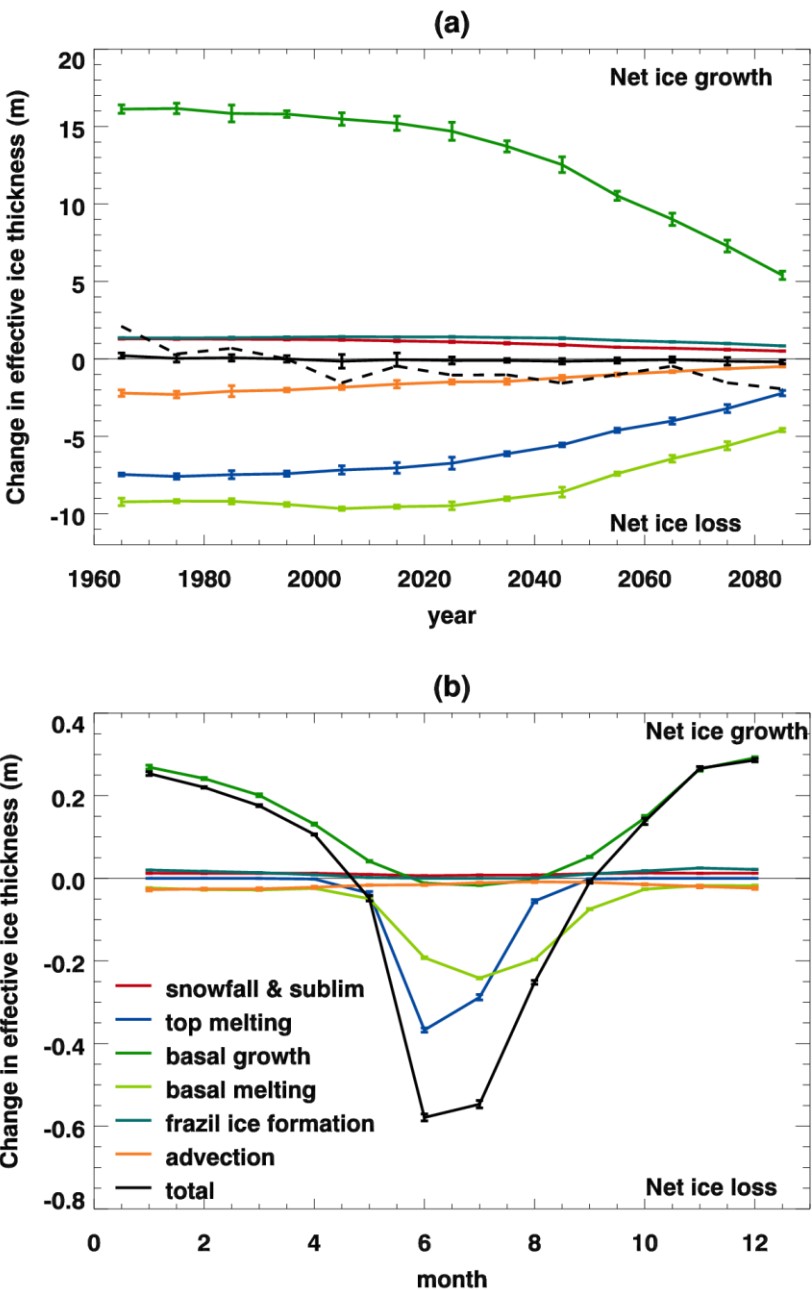

**Figure 4:** Components of the sea ice volume budget as defined in section 3.1 for the HadGEM2-ES Hist+RCP8.5 integrations, averaged over the region defined in Fig. 2. Values are ensemble means +/- 1 standard deviation, and positive values correspond to net ice growth.

5   (a) Decadal mean timeseries, with the dash line showing the sum of the budget terms magnified by a factor of 10.

  (b) Seasonal cycle for the reference period 1960-89.

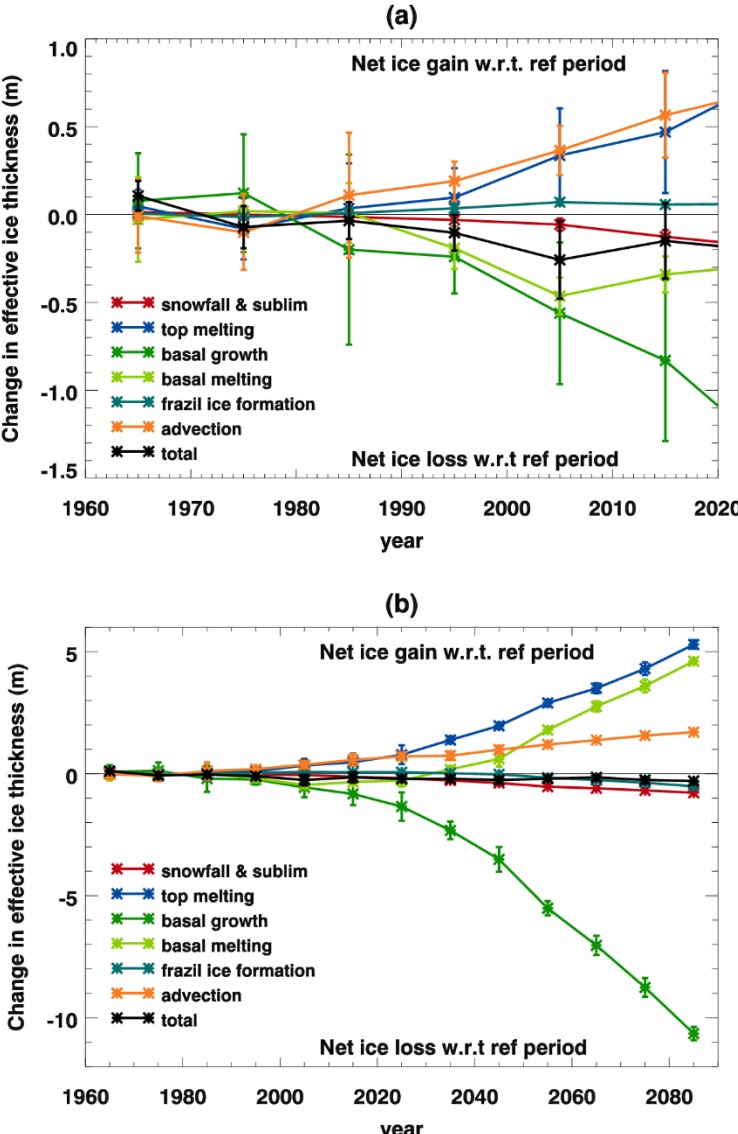

**Figure 5:** Decadal mean components of the sea ice volume budget as defined in section 3.1 for the HadGEM2-ES Hist+RCP8.5 integrations, averaged over the region defined in Fig. 2 and plotted as differences relative to the mean over the reference period 1960-89. Values are ensemble means +/- 1 standard deviation, and positive values correspond to net ice gain w.r.t. the reference period.

(a) To 2020 (with magnified vertical scale)   (b) To 2090

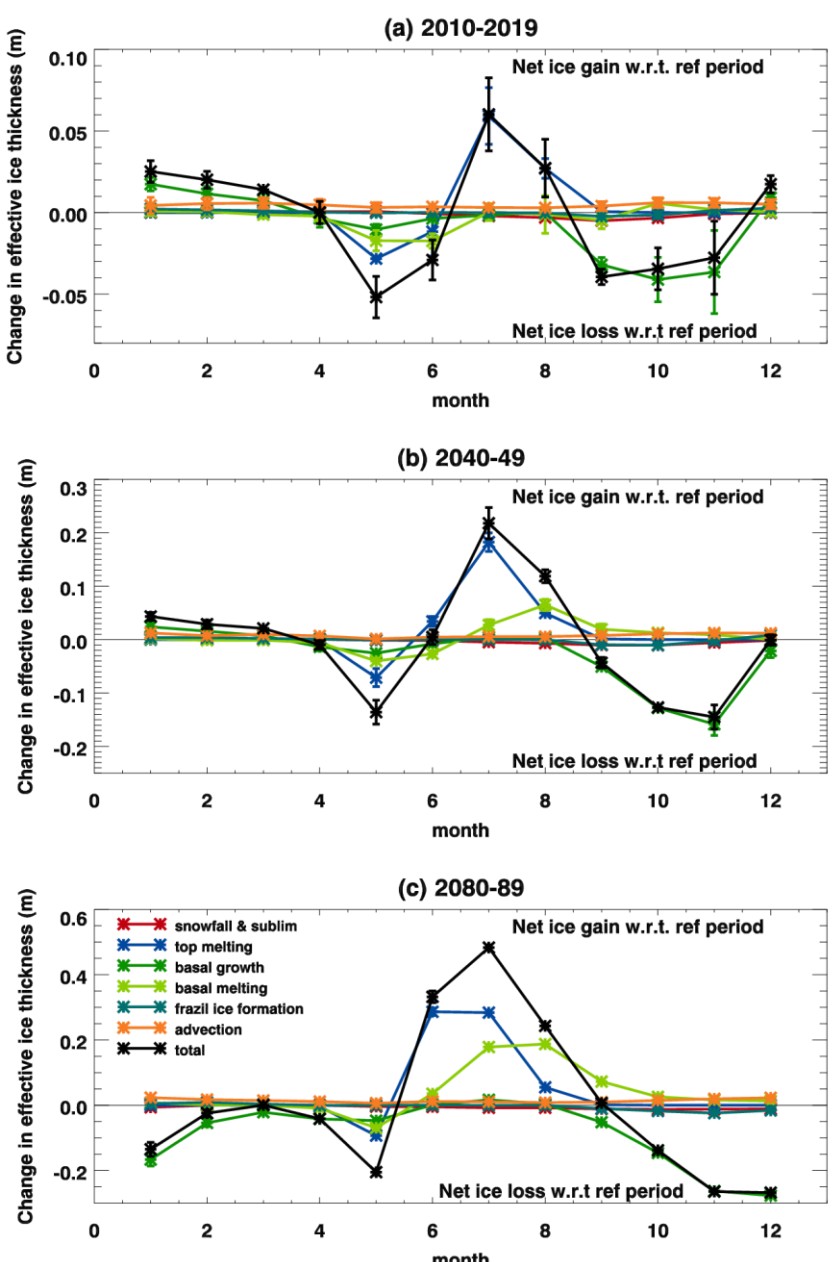

**Figure 6:** Ensemble mean seasonal cycles of the sea ice volume budget components as defined in section 3.1 for the HadGEM2-ES Hist+RCP8.5 integrations, averaged over the region defined in Fig. 2 and plotted as differences relative to the mean over the reference period 1960-89. Values are ensemble means +/- 1 standard deviation, and positive values correspond to net ice gain w.r.t. the reference period. (a) 2010-2019 (b) 2040-49 (c) 2080-89. Note that the plots have different vertical scales.

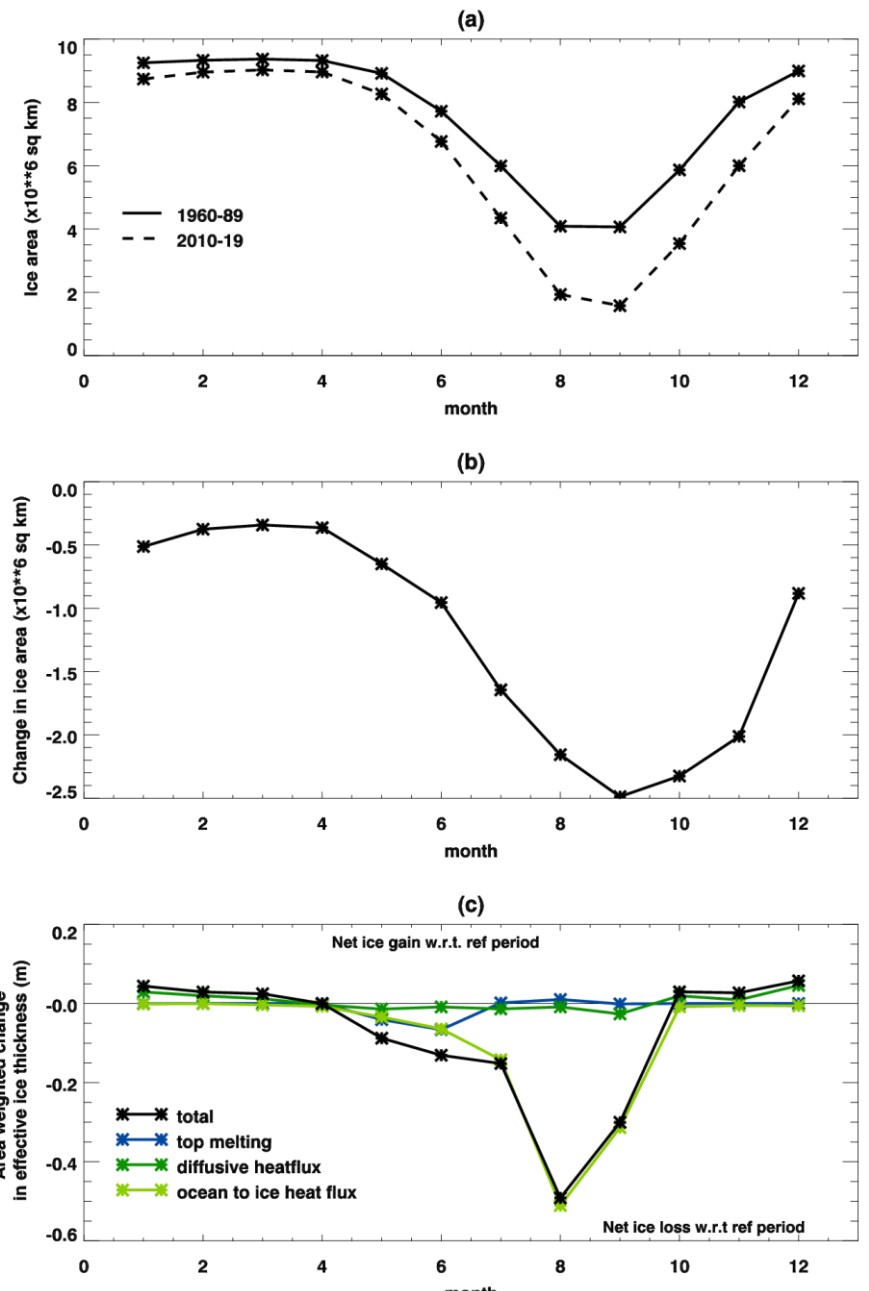

**Figure 7:** (a) Ice area over the domain defined in Fig. 2 during the reference period 1960-89 (solid lines) and 2010-19 (broken lines) for the HadGEM2-ES Hist1+RCP8.5 scenario. (b) Change in ice area between 1960-89 and 2010-19. (c) Seasonal cycles of changes in selected sea ice volume budget components for 2010-19 w.r.t. the reference period 1960-89 for the HadGEM2-ES Hist1+RCP8.5 integration. The components are defined in section 3.1. Values are averaged over the region defined in Fig. 2 and weighted by the ice area in each case, so that the change is per unit area of the remaining ice.

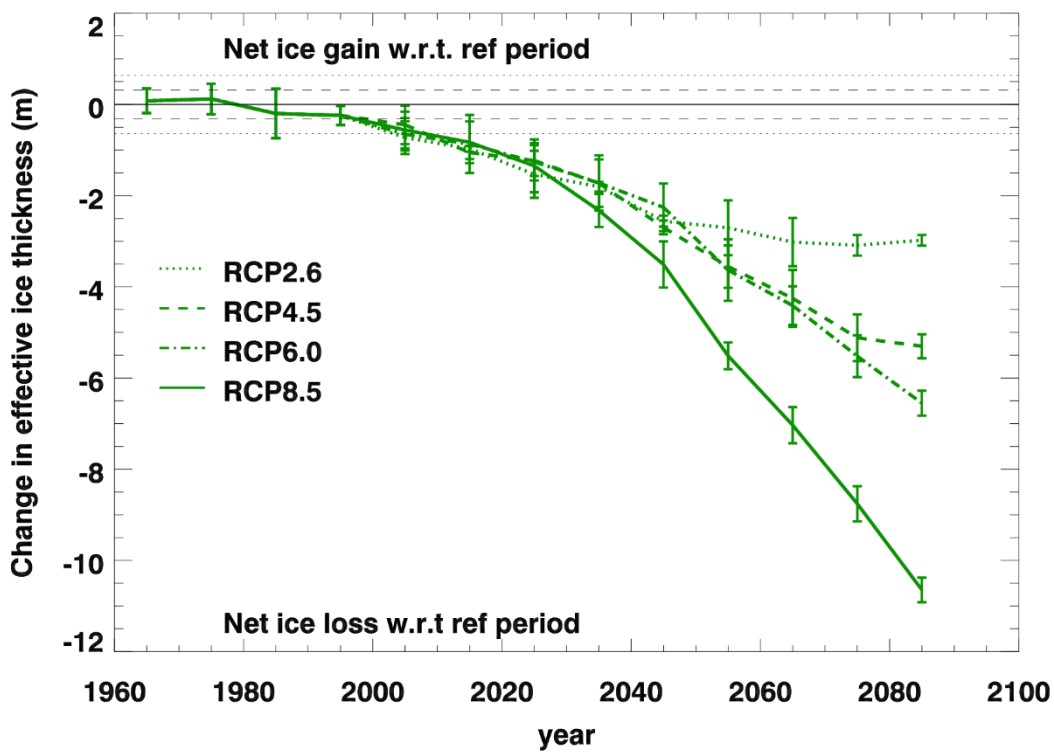

**Figure 8:** Decadal mean values of the basal growth component of the sea ice volume budget of HadGEM2-ES, plotted as differences relative to the reference period 1960-89, for each of the forcing scenarios. Values are ensemble means +/- 1 standard deviation, and positive values correspond to net ice gain w.r.t. the reference period. The horizontal dashed and dotted lines show +/- 1 and 2 standard deviations as calculated from 250 years of the HadGEM2-ES pre-industrial control integration.

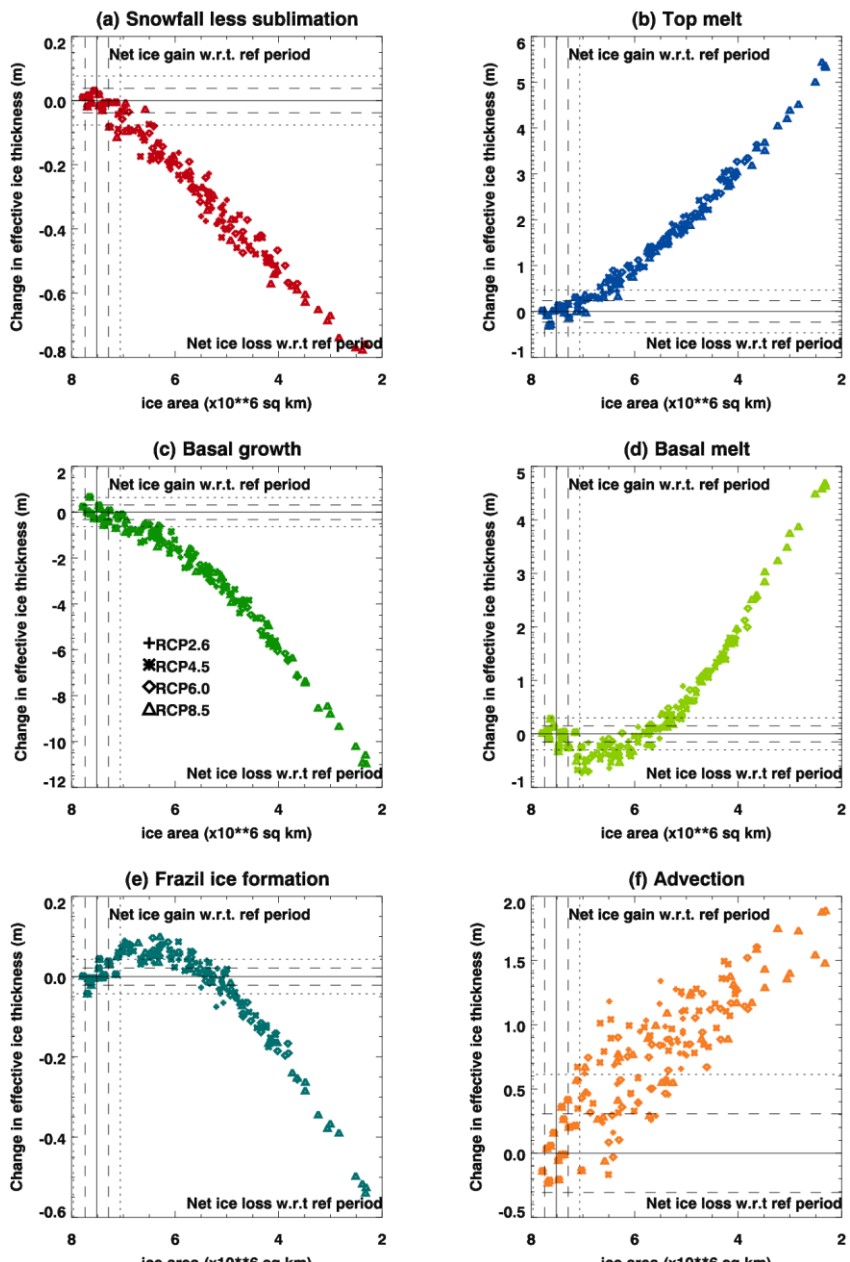

**Figure 9:** Decadal mean HadGEM2-ES sea ice volume budget components for each ensemble member and for all the forcing scenarios, plotted as differences relative to the reference period 1960-89, and as a function of the decadal mean ice area. Positive values correspond to net ice gain relative to the reference period. The horizontal and vertical dashed and dotted lines show +/- 1 and 2 standard deviations as calculated from 250 years of the HadGEM2-ES pre-industrial integration. Note that the plots have different vertical scales.

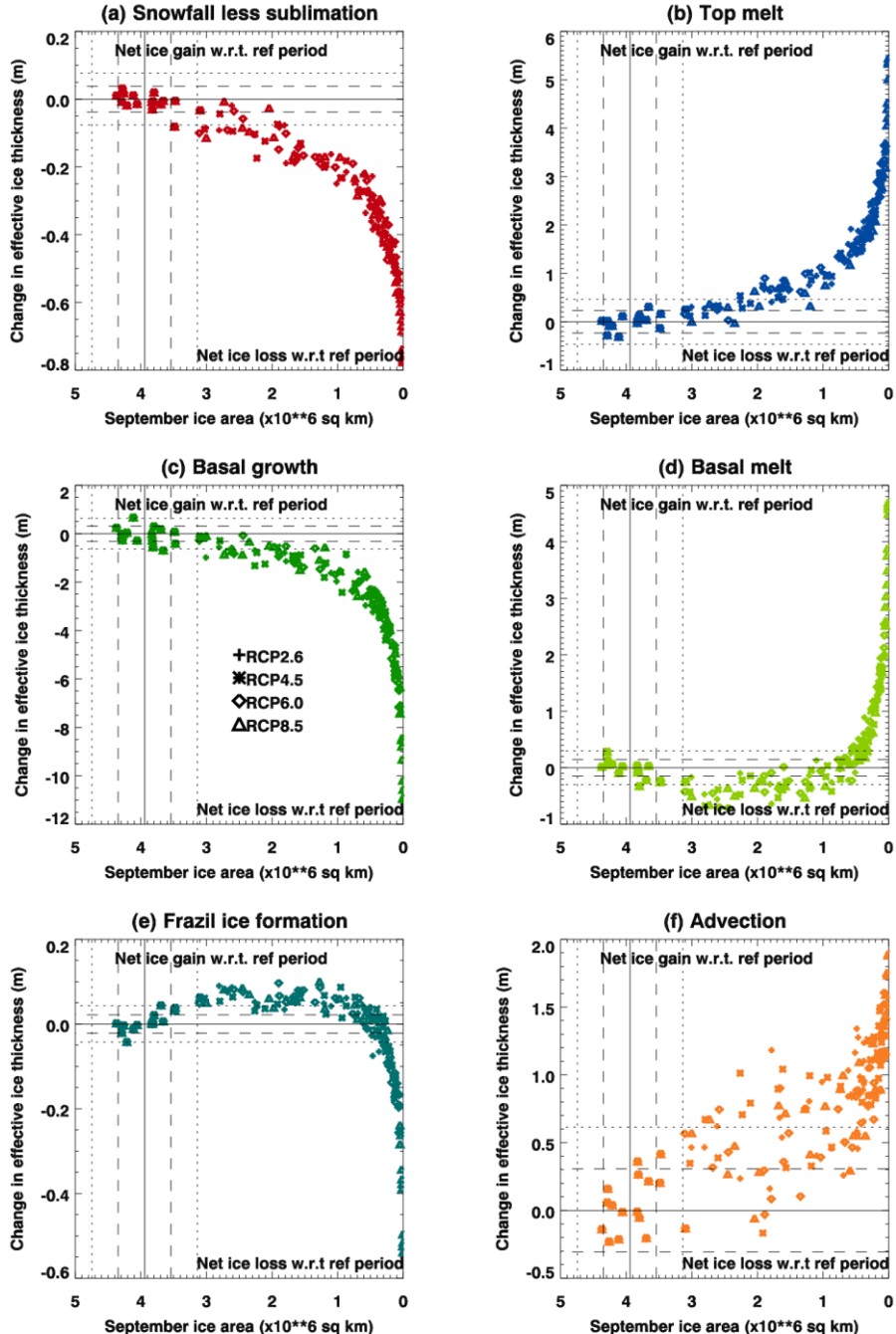

**Figure 10:** As figure 9, but plotted against decadal mean September ice area rather than the decadal mean over all months.