# Peer review of "Investigating future changes in the volume budget of the Arctic sea ice in a coupled climate model"

_The Cryosphere, 2017_

## Referee Comment (RC1) · Anonymous Referee #1 · 30 Nov 2017

Summary

The authors investigate changes in the Arctic sea-ice volume during the 21st century. To do so, they use the Earth System Model HadGEM2-ES and output variables that describe the different components of the ice volume budget, i.e. basal melting and growth, top melting, snowfall, frazil ice formation, and ice advection. The effects of these processes on the ice volume can directly be quantified as they can be transformed into meters of ice thickness. Therefore the ice volume budget can be closed. The method enables a thorough analysis of the evolution of the sea-ice volume budget during the 21st century. The authors find that the sea-ice loss is mainly driven by a decrease in

basal growth over the 21st century in the decadal mean. However, by investigating the seasonal cycle, they show that different processes are at work depending on the time of the year. Finally, another important result of the study is that the changes in the processes do not depend on the forcing of the scenario but rather on the sea-ice area that is still present.

As there is still a high spread in climate model projections, this topic is interesting and could bring more insight into differences in ice volume budget evolution in different models. This method could be used for comparison between CMIP6 models, if these provide the needed variables, as suggested by the SIMIP protocol. Therefore, the topic is of relevance in the current context of sea-ice and climate research. The manuscript is well written but I would appreciate if the authors would clarify some points. I also have some additional suggestions.

Thematic comments

**1 In the end of the introduction, it is not clear to the reader what precisely is the scope of the study and what is new about it. It is clear that the authors will describe the evolution of the sea-ice volume budget, in a similar way to Holland et al. (2010), with the method of Keen et al. (2013). However, it is not clear if the scope of the manuscript is to introduce the method for further application (as is suggested in the conclusion) or to draw conclusions from the ice volume budget evolution to improve the understanding of changes in the Arctic climate system as a whole. I would appreciate if the authors make this point very clear in the beginning. It is difficult to follow the story of the manuscript otherwise. The authors write on page 5: "although here we are also able to include individual components [. . .] volume budget". I suggest including this information in the introduction as it is a strong statement about what makes this study special.**

**2 The reference period is chosen as the years 1960-79. I would like the authors to comment why they chose this period and not the period 1960-1989 (as usually used in**

studies for IPCC assessment reports) or the period 1950-79 (to have at least 30 years). I find this period rather short to be a reference period. I wonder if they have tried other reference periods? And if yes, do they yield different results?

**3 P4 L22-24: I do not understand why Eq. 1 should result in an ice volume that has to be converted back to effective ice thickness. As far as I understood, Eq. 1 gives thicknesses directly. I would appreciate if the authors could clarify this.**

**4 P4 L30: Can the authors explain the especially steep decline of the winter sea-ice cover from 2080 onwards in the RCP8.5 scenario with their results? I would guess it has to do with the increase in water temperature inhibiting the formation of a winter sea-ice cover (see Bathiany et al. 2016). I would find interesting to hear if the authors have another explanation. It would be worth mentioning in the manuscript as well. On the same note, I would suggest that the greater decline in basal ice growth (P9 L26) is linked to the greater decline in ice area in RCP8.5 stated earlier in the manuscript. Maybe these two could be linked to make a statement about the processes at work here.**

**5 P7 L4: The lateral melting is not explicitly modeled. Do the authors have an idea of how important this term is? I could imagine that it is an important term in summer, as a component of the sea-ice albedo feedback.**

**6 P7 L19-22: Holland found large differences between CMIP3 models. I would like the authors to comment on the implications of their findings for these differences or differences between CMIP5 models. In any case, I suggest moving this paragraph to the discussion in the end of the manuscript.**

**7 P8 L29: The authors write that the extra top melting is enhanced by reductions in the surface albedo. Do they infer this directly from the model simulation? I wonder if maybe longwave radiation also has an influence on surface melting (see e.g. Notz and Stroeve, 2016), for example through clouds and water vapor? I would like the authors to comment on that.**

[Figure]

**8 P9 L2: I am not convinced that the in-situ warming of the ocean is only a consequence of the ice cover retreat in your model. Could it not also partly be due to a higher advection of oceanic heat from lower latitudes, as stated for example in Burgard and Notz, 2017? I would like the authors to comment on that.**

**9 The conclusion from Fig. 9 and Fig. 10 is that the changes in components of the ice volume budget are independent of the forcing and dependent on the remaining sea-ice area. I agree that this relationship is very clear. However, can the authors be sure that it is not rather dependent on the temperature? Several studies showed that the sea-ice area depends linearly on the air temperature (e.g. Winton, 2011; Mahlstein and Knutti, 2012) and cumulated $CO_2$ emissions (Notz and Stroeve, 2016). It might be worth having a look at these relationships as well to get a larger picture and maybe a stronger conclusion.**

**10 The last paragraph of the conclusion is somewhat unclear and is not very strong. This is not an advantage for the manuscript. I would suggest discussing a little more what makes this study special and what are its implications for future research. It is still not clear enough for me.**

Writing comments

**11 The Section 2.2 about model integration is interesting but I think there are too many details. The effect of the different $CO_2$ pathways on the temperature is what is important for the study. This effect can be seen well in Fig. 1. I therefore suggest that the authors leave in the reference to Moss et al. (2010) but that they leave out the bullet point list and the sentence "Fig. 1A of Caesar et al. [. . .] scenarios."**

**12 I suggest writing down the exact limits for the study area in an appendix/supporting information. This might be useful for the comparison with future studies.**

**13 Section 3.1, P5-6: The bullet point list makes the text well readable. To keep consistent, maybe the authors could add some numbers to the three first points. There,**

the results are described qualitatively in contrast to the three last points, where they are described quantitatively.

**14 The transition between Section 3 and Section 4 is quite abrupt. I would suggest working on a more logical transition.**

**15 In section 4.1., Fig. 5B is cited instead of Fig. 5A and vice versa. I suggest reading through this section carefully again.**

**16 In section 4.2., the reader is pointed to several different figures while the rest of the manuscript is very structured (one paragraph = one figure description). In this case, it is helpful for the message to look at the different figures. However, I find difficult to follow the story from P9 L1 to P9 L22. I suggest to try reformulating the message in a clearer way.**

**17 P10 L26-27: The processes changing at the ice surface are listed and then "basal melting" is mentioned. Why?**

Technical comments

P1 L9-13: These two sentences are long and contain too much information. Reformulating might clarify the message.

P3 L3: I suggest removing "for use in IPCC AR5". I think readers know the aim of CMIP5.

P3 L9: West et al., 2017 is cited. In the references, it is marked as "in prep.". I think they can therefore not be cited it in this context then.

P3 L12: Replace "as that used" by "as the one used"

P4 L6: Remove comma after the Moss et al., 2010 reference

P4 L15-16: The sentence is long. I suggest cutting after "scenario)" and starting the next sentence with "Fig.1".

P6 L25: The sentence is too long. I suggest stopping after "loss" and starting the next sentence with "The ice decline arises"

P6 L27: Add "seen in Fig. 3b" after "thickness".

P6 L29: The sentence is long. I suggest stopping after "line." and starting next sentence with "During"

P7 L8: Replace "and also how the seasonal cycle changes" by "and the changes in seasonal cycle".

P7 L27: "s" missing after 2040

P8 L 24-26: This sentence is too long. I suggest reformulating it to clarify the message.

P8 L31-32: Can the authors reformulate this sentence? I do not understand it.

P9 L9: add "process" between "this" and "that"

P11 L2: I suggest changing "and reduced basal growth during autumn/early winter" to "and in autumn/early winter due to reduced basal growth".

P11 L15-18: This sentence is too long and unclear. I suggest reformulating it to clarify the message.

Figures

Fig. 1: I suggest marking or shading the reference period In the caption, replace "HadGEM2ES" by "HadGEM2-ES"

Fig. 6: Have the authors looked into the period 2080-2099? Are the changes still similar? If not, would they bring additional information for the study?

Fig. 7: Add in the caption for (b) that this is 2010-2019.

Fig. 9: 1960-79 instead of 1960-9

References

Arctic is written "arctic" in most of your references. I suggest reading through them carefully again.

Please also note the supplement to this comment:
https://www.the-cryosphere-discuss.net/tc-2017-216/tc-2017-216-RC1-supplement.pdf
* * *

---

## Referee Comment (RC2) · Anonymous Referee #2 · 11 Dec 2017

General Comments

In this work the authors decompose in the coupled climate model HadGEM2-ES the global Arctic sea ice volume budget over the late 20th and 21st century into its main components – top melt, basal growth, basal melt, frazil ice formation, advection, snowfall less sublimation.

In many ways this study appears as a follow up study of the earlier Keen et al., 2013 paper - see section 5 on 'modelled heat budget of the Arctic snow and ice' but instead of taking a local (per unit ice area) analysis here the authors present a global perspective that presents the advantage of explaining the mechanisms that control sea ice volume

decline at the Arctic basin scale.

The main results of this study are:

- To present a detailed methodology of how to analyse the HadGEM2-ES Arctic sea ice volume budget components at the basin scale - To characterise and rank in order of importance the different terms controlling the seasonal and inter-annual sea ice growth (and melt) - To show that the changes in the volume budget are a function of the sea ice cover and not of the speed at which the sea ice retreats

My overall impression is that there is nothing fundamentally wrong with this paper but that at the same time that it does not contribute to any significant advances in the field.

I encourage the authors to explore one of the following possible extensions of their work in order to give it a wider audience:

- Explore impact of sea ice physics even at a simple level. Comparing results with results from HadGEM1 analyzed in Keen et al., 2013 could be informative. While it would be difficult to separate the impact of the different physics in the two models on the total volume budget it would show how model developments modify our understanding of the drivers of sea ice decline. - Compare the model results with other climate models. In that sense the reader would get a better sense of inter-model variability. The authors suggest that their methodology is appropriate to analyse other models. Why not do it? - If these options appear too ambitious the authors may at least consider improving the quality of the figures and explain in greater details how the decompositions presented in those figures help explain the future evolution of the sea ice cover and its role in the climate as a whole. For example what can we learn about the changing climate based on seasonal changes in the different terms in the volume budget. Similarly what do the figures 9 and 10 on the changes of effective thickness as a function of sea ice area tells us about climate change in the Arctic and beyond.

---

## Author Comment (AC1) · 14 Mar 2018

We thank the reviewer for reading our manuscript, and for his/her comments. Our responses are included below in blue text.

Summary

The authors investigate changes in the Arctic sea-ice volume during the 21st century. To do so, they use the Earth System Model HadGEM2-ES and output variables that describe the different components of the ice volume budget, i.e. basal melting and growth, top melting, snowfall, frazil ice formation, and ice advection. The effects of these processes on the ice volume can directly be quantified as they can be transformed into meters of ice thickness. Therefore the ice volume budget can be closed. The method enables a thorough analysis of the evolution of the sea-ice volume budget during the 21st century. The authors find that the sea-ice loss is mainly driven by a decrease in basal growth over the 21st century in the decadal mean. However, by investigating the seasonal cycle, they show that different processes are at work depending on the time of the year. Finally, another important result of the study is that the changes in the processes do not depend on the forcing of the scenario but rather on the sea-ice area that is still present.

As there is still a high spread in climate model projections, this topic is interesting and could bring more insight into differences in ice volume budget evolution in different models. This method could be used for comparison between CMIP6 models, if these provide the needed variables, as suggested by the SIMIP protocol. Therefore, the topic is of relevance in the current context of sea-ice and climate research. The manuscript is well written but I would appreciate if the authors would clarify some points. I also have some additional suggestions.

Thematic comments

**1 In the end of the introduction, it is not clear to the reader what precisely is the scope of the study and what is new about it. It is clear that the authors will describe the evolution of the sea-ice volume budget, in a similar way to Holland et al. (2010), with the method of Keen et al. (2013). However, it is not clear if the scope of the manuscript is to introduce the method for further application (as is suggested in the conclusion) or to draw conclusions from the ice volume budget evolution to improve the understanding of changes in the Arctic climate system as a whole. I would appreciate if the authors make this point very clear in the beginning. It is difficult to follow the story of the manuscript otherwise. The authors write on page 5: "although here we are also able to include individual components [. . .] volume budget". I suggest including this information in the introduction as it is a strong statement about what makes this study special.**

We intend the scope of the manuscript to be both the introduction of the method, illustrated by the application to the HadGEM2-ES model, and the investigation of the impact of the forcing scenario on the budget changes. This will be stated more explicitly in the revised manuscript and the statement on page 5 will be included in the introduction as suggested.

**2 The reference period is chosen as the years 1960-79. I would like the authors to comment why they chose this period and not the period 1960-1989 (as usually used in studies for IPCC assessment reports) or the period 1950-79 (to have at least 30 years).**
I find this period rather short to be a reference period. I wonder if they have tried other reference periods? And if yes, do they yield different results?

We have also considered changes w.r.t the 30 year period 1960-89, and the results and conclusions are almost identical. In the revised manuscript we will use this 30 year period for reference to avoid any concern about the period being too short.

**3 P4 L22-24: I do not understand why Eq. 1 should result in an ice volume that has to be converted back to effective ice thickness. As far as I understood, Eq. 1 gives thicknesses directly. I would appreciate if the authors could clarify this.**

Apologies, this is incorrect and this paragraph will read as follows in the revised manuscript:
We focus on changes in the sea ice over the domain shown in Fig. 2, covering the Arctic basin, and the Barents Sea. Figure 3 shows how the ice area and mean effective ice thickness within this domain declines for each of the model integrations during the period 1960 to 2090. The effective ice thickness includes the impact of any overlying snow by converting the snow to an equivalent thickness of ice using Eq. (1). Hereafter, whenever ice thickness or volume is mentioned it refers to an *effective* value, which includes the overlying snow as well.

**4 P4 L30: Can the authors explain the especially steep decline of the winter sea-ice cover from 2080 onwards in the RCP8.5 scenario with their results? I would guess it has to do with the increase in water temperature inhibiting the formation of a winter sea-ice cover (see Bathiany et al. 2016). I would find interesting to hear if the authors have another explanation. It would be worth mentioning in the manuscript as well. On the same note, I would suggest that the greater decline in basal ice growth (P9 L26) is linked to the greater decline in ice area in RCP8.5 stated earlier in the manuscript. Maybe these two could be linked to make a statement about the processes at work here.**

Yes, it is most likely that the steeper decline in winter ice cover towards the end of the 21$^{st}$ century in RCP8.5 is associated with the warming ocean surface: certainly the DJF ocean top level temperature increases more rapidly towards the end of the integration. The link between the declining basal ice growth and the declining ice area is shown in Figures 9 and 10, but we agree that the faster decline towards the end of the 21$^{st}$ century could be discussed more explicitly, and linked with the ocean changes, and this will be done in the revised manuscript.

**5 P7 L4: The lateral melting is not explicitly modeled. Do the authors have an idea of how important this term is? I could imagine that it is an important term in summer, as a component of the sea-ice albedo feedback.**

In a 'present day' (year 2000) long equilibrium run of our latest (CMIP6) model HadGEM3 GC3.1, the lateral melting term is important in the mean budget of the Arctic sea ice during JJA, when it is at most about 14% of the ocean to ice heat flux. It may become more important in a warming climate, and while this is outside the scope of this manuscript it will be possible to investigate this using data from CMIP6 scenario runs once they are available.
HadGEM2-ES does include an adjustment [*] to the ocean to ice heat flux when the ice concentration drops below 0.05, to provide a crude representation of increased lateral melting of small ice floes in the marginal ice zone.
[*] The heat flux is scaled by 0.05/ice_area so that the grid box integral of the flux becomes independent of the ice area .

In the revised manuscript we will include more about the lateral melting in the discussion.

**6 P7 L19-22: Holland found large differences between CMIP3 models. I would like the authors to comment on the implications of their findings for these differences or differences between CMIP5 models. In any case, I suggest moving this paragraph to the discussion in the end of the manuscript.**

The point we intended to make here was that given Holland et al found that CMIP3 models did not even agree on the relative role of melt and growth in the ice decline, we might also expect considerable inter-model differences when we can break these terms down further for the CMIP6 models. We agree that it would be better to move this to the discussion, and will clarify our text.

**7 P8 L29: The authors write that the extra top melting is enhanced by reductions in the surface albedo. Do they infer this directly from the model simulation? I wonder if maybe longwave radiation also has an influence on surface melting (see e.g. Notz and**

Stroeve, 2016), for example through clouds and water vapor? I would like the authors
to comment on that.

Yes we can see reductions in the surface albedo directly in the model. We now plan to look at the surface fluxes in more detail and provide a more comprehensive discussion of the factors contributing to changes in the surface melting.

**8 P9 L2: I am not convinced that the in-situ warming of the ocean is only a consequence of the ice cover retreat in your model. Could it not also partly be due to a higher advection of oceanic heat from lower latitudes, as stated for example in Burgard and Notz, 2017? I would like the authors to comment on that.**

Yes, the advection of oceanic heat from lower latitudes does also play a role, and as shown in Burgard and Notz it is the main driver for the long term warming of the Arctic Ocean. However a seasonal analysis of the Arctic Ocean budget for HadGEM2-ES shows that during the spring (MAM) and summer (JJA), when large increases in the basal melting are seen, atmospheric surface fluxes are the major driver of warming, especially for the upper ocean. We will expand the discussion here to more fully describe the causes of ocean warming.

**9 The conclusion from Fig. 9 and Fig. 10 is that the changes in components of the ice volume budget are independent of the forcing and dependent on the remaining sea-ice area. I agree that this relationship is very clear. However, can the authors be sure that it is not rather dependent on the temperature? Several studies showed that the sea-ice area depends linearly on the air temperature (e.g. Winton, 2011; Mahlstein and Knutti, 2012) and cumulated CO2 emissions (Notz and Stroeve, 2016). It might be worth having a look at these relationships as well to get a larger picture and maybe a stronger conclusion.**

Yes, there is a linear relationship between anomalies in the ice area and the near-surface air temperature in HadGEM2-ES, which holds for all the forcing scenarios considered here. We did consider plotting Fig 9 and Fig 10 using temperature instead of ice area. The main reason that we decided not to was because we felt that the changes in the budget terms and the changes in ice area were more directly and closely linked (eg smaller basal growth term in the ice budget because the growth is occurring over a smaller area). We agree that it would be advantageous to widen the discussion to include these relationships as well and we will do this, and mention that the relationships shown in Fig. 9 and Fig. 10 are similar for the near-surface temperature.

**10 The last paragraph of the conclusion is somewhat unclear and is not very strong. This is not an advantage for the manuscript. I would suggest discussing a little more what makes this study special and what are its implications for future research. It is still not clear enough for me.**

We will expand the summary and discussion section to clarify what we feel are the key novel aspects of this study, which are:
- Our formulation of the volume budget includes individual components of the melt/freeze terms, so we know whether changes are attributable to atmospheric or oceanic processes.
- We consider the seasonal cycle of these changes, to understand the (sometimes opposing) changes at different times of year.
- We consider different ways of looking at changes in the budget components:
  - Ensuring the (declining) ice area is taken into account in order to construct a budget that balances the changing ice volume, and
    - Considering 'local' changes over the ice itself, which are more easily related to physical processes (eg more surface melting in a warmer climate)
  The combination of these approaches helps to understand the (important) impact of the declining ice cover on the budget.

In terms of future research, this approach can be applied to the CMIP6 models to gain a greater understanding of the process changes behind the modelled decline in ice area and volume, and we intend to do this once the CMIP6 data becomes available.

Writing comments
**11 The Section 2.2 about model integration is interesting but I think there are too**

many details. The effect of the different CO2 pathways on the temperature is what is important for the study. This effect can be seen well in Fig. 1. I therefore suggest that the authors leave in the reference to Moss et al. (2010) but that they leave out the bullet point list and the sentence "Fig. 1A of Caesar et al. [. . .] scenarios."

**12 I suggest writing down the exact limits for the study area in an appendix/supporting information. This might be useful for the comparison with future studies.**

**13 Section 3.1, P5-6: The bullet point list makes the text well readable. To keep consistent, maybe the authors could add some numbers to the three first points. There, the results are described qualitatively in contrast to the three last points, where they are described quantitatively.**

The above suggestions will be incorporated into the revised manuscript

**14 The transition between Section 3 and Section 4 is quite abrupt. I would suggest working on a more logical transition.**

Yes, agreed, some linking text will be added to improve the flow here.

**15 In section 4.1., Fig. 5B is cited instead of Fig. 5A and vice versa. I suggest reading through this section carefully again.**

Thank you – yes this will be checked and corrected.

**16 In section 4.2., the reader is pointed to several different figures while the rest of the manuscript is very structured (one paragraph = one figure description). In this case, it is helpful for the message to look at the different figures. However, I find difficult to follow the story from P9 L1 to P9 L22. I suggest to try reformulating the message in a clearer way.**

This section will be redrafted.

**17 P10 L26-27: The processes changing at the ice surface are listed and then "basal melting" is mentioned. Why?**

Apologies, this was not intended to refer to processes acting at the top surface of the ice, and should read: For this model, the processes that change *most per unit area of the ice* as the climate warms are......

Technical comments
P1 L9-13: These two sentences are long and contain too much information. Reformulating might clarify the message.
P3 L3: I suggest removing "for use in IPCC AR5". I think readers know the aim of CMIP5.
P3 L9: West et al., 2017 is cited. In the references, it is marked as "in prep.". I think they can therefore not be cited it in this context then.
P3 L12: Replace "as that used" by "as the one used"
P4 L6: Remove comma after the Moss et al., 2010 reference
P4 L15-16: The sentence is long. I suggest cutting after "scenario)" and starting the next sentence with "Fig.1".
P6 L25: The sentence is too long. I suggest stopping after "loss" and starting the next sentence with "The ice decline arises"
P6 L27: Add "seen in Fig. 3b" after "thickness".
P6 L29: The sentence is long. I suggest stopping after "line." and starting next sentence with "During"
P7 L8: Replace "and also how the seasonal cycle changes" by "and the changes in seasonal cycle".
P7 L27: "s" missing after 2040
P8 L 24-26: This sentence is too long. I suggest reformulating it to clarify the message.

The changes suggested above will be incorporated into the revised manuscript.

P8 L31-32: Can the authors reformulate this sentence? I do not understand it.

The revised manuscript will contain a reformulated version, for example:
So during July and August, the amount of top melting per unit area of the ice is about the same during 2010-19 and the reference period (Fig. 7). Since the ice cover is lower in 2010-19, this results in a smaller volume of ice melt, expressed in Figure 6 as a net ice gain w.r.t the reference period. This same effect is also seen during later decades.

P9 L9: add "process" between "this" and "that"
P11 L2: I suggest changing "and reduced basal growth during autumn/early winter" to "and in autumn/early winter due to reduced basal growth".

The changes suggested above will be incorporated into the revised manuscript.

P11 L15-18: This sentence is too long and unclear. I suggest reformulating it to clarify the message.

This whole paragraph will be rewritten to address comment #10 above.

Figures
Fig. 1: I suggest marking or shading the reference period In the caption, replace "HadGEM2ES" by "HadGEM2-ES"
Fig. 6: Have the authors looked into the period 2080-2099? Are the changes still similar? If not, would they bring additional information for the study?
Fig. 7: Add in the caption for (b) that this is 2010-2019.
Fig. 9: 1960-79 instead of 1960-9

The suggested changes to Figures 1,7 and 9 will be made.

Regarding Figure 6, yes we looked at these changes for every decade to the end of the 21st century, and for each of the forcing scenarios. We did consider including a third decade, later in the scenario, in figure 6 (we had included 2070-79 in this figure in earlier drafts of the manuscript). However we decided not to include it because the general signals of change seen in these seasonal cycles plots were similar and did not seem to add to the discussion, and we felt that the impact of the Arctic becoming seasonally ice free was better illustrated in figures 9 and 10. In the revised manuscript we will mention that the changes in the later decades are similar.

References
Arctic is written "arctic" in most of your references. I suggest reading through them carefully again.

The references will be checked and corrected where necessary.

---

## Author Comment (AC2) · 14 Mar 2018

We thank the reviewer for reading our manuscript, and for his/her comments. Our responses are included below in blue text.

General Comments
In this work the authors decompose in the coupled climate model HadGEM2-ES the global Arctic sea ice volume budget over the late 20th and 21st century into its main components – top melt, basal growth, basal melt, frazil ice formation, advection, snowfall less sublimation.
In many ways this study appears as a follow up study of the earlier Keen et al., 2013 paper - see section 5 on 'modelled heat budget of the Arctic snow and ice' but instead of taking a local (per unit ice area) analysis here the authors present a global perspective that presents the advantage of explaining the mechanisms that control sea ice volume decline at the Arctic basin scale.
The main results of this study are:
- To present a detailed methodology of how to analyse the HadGEM2-ES Arctic sea ice volume budget components at the basin scale - To characterise and rank in order of importance the different terms controlling the seasonal and inter-annual sea ice growth (and melt) - To show that the changes in the volume budget are a function of the sea ice cover and not of the speed at which the sea ice retreats

My overall impression is that there is nothing fundamentally wrong with this paper but that at the same time that it does not contribute to any significant advances in the field.

We feel that the type of study outlined here is now necessary in order to understand inter-model differences in projected future ice decline: we need to consider changes in the underlying processes as well as looking at how the ice state changes. We also feel that there are a number of novel aspects to our study, some of which are summarised below. However we recognise from this review, and from comments from reviewer 1, that we have not been clear enough about the scope and interest of the study so we will be updating the manuscript to improve this.

- Our formulation of the volume budget includes individual components of the melt/freeze terms, so we know whether changes are attributable to atmospheric or oceanic processes.
- We consider the seasonal cycle of these changes, to understand the (sometimes opposing) changes at different times of year.
- We consider different ways of looking at changes in the budget components:
    - Ensuring the (declining) ice area is taken into account in order to construct a budget that balances the changing ice volume, and
    - Considering 'local' changes over the ice itself, which are more easily related to physical processes (eg more surface melting in a warmer climate)
  The combination of these approaches helps to understand the (important) impact of the declining ice cover on the budget.

I encourage the authors to explore one of the following possible extensions of their work in order to give it a wider audience:

- Explore impact of sea ice physics even at a simple level. Comparing results with results from HadGEM1 analyzed in Keen et al., 2013 could be informative. While it would be difficult to separate the impact of the different physics in the two models on the total volume budget it would show how model developments modify our understanding of the drivers of sea ice decline

We have applied the same analysis to HadGEM1, but the sea ice physics in these two models is essentially the same, so this comparison would not show the impact of different sea ice physics. The two

models showed very similar budget change, with the main differences being due to different rates of ice decline. While we *are* interested in the impact of the ice decline in the changing budget, we felt this was better illustrated using one model and a range of different forcing scenarios.

. - Compare the model results with other climate models.
In that sense the reader would get a better sense of inter-model variability. The authors suggest that their methodology is appropriate to analyse other models. Why not do it?

This will be possible using the diagnostics from CMIP6 models, and we plan to do this once the data is available.

- If these options appear too ambitious the authors may at least consider improving the quality of the figures and explain in greater details how the decompositions presented in those figures help explain the future evolution of the sea ice cover and its role in the climate as a whole. For example what can we learn about the changing climate based on seasonal changes in the different terms in the volume budget. Similarly what do the figures 9 and 10 on the changes of effective thickness as a function of sea ice area tells us about climate change in the Arctic and beyond.

We will be widening the discussion to include:
- Drivers of the ocean warming causing extra basal melt
- Relationship between the ice area and global changes
- Changes in the surface fluxes contributing to melt at the ice surface.

---

## Author Response (AR1)

We thank the reviewer for reading our manuscript, and for his/her comments. Our updated responses are included below in blue text. The page numbers refer to the 'tracked-changes' section of this document, showing the alterations from the original version of the manuscript.

Summary

The authors investigate changes in the Arctic sea-ice volume during the 21st century. To do so, they use the Earth System Model HadGEM2-ES and output variables that describe the different components of the ice volume budget, i.e. basal melting and growth, top melting, snowfall, frazil ice formation, and ice advection. The effects of these processes on the ice volume can directly be quantified as they can be transformed into meters of ice thickness. Therefore the ice volume budget can be closed. The method enables a thorough analysis of the evolution of the sea-ice volume budget during the 21st century. The authors find that the sea-ice loss is mainly driven by a decrease in basal growth over the 21st century in the decadal mean. However, by investigating the seasonal cycle, they show that different processes are at work depending on the time of the year. Finally, another important result of the study is that the changes in the processes do not depend on the forcing of the scenario but rather on the sea-ice area that is still present.

As there is still a high spread in climate model projections, this topic is interesting and could bring more insight into differences in ice volume budget evolution in different models. This method could be used for comparison between CMIP6 models, if these provide the needed variables, as suggested by the SIMIP protocol. Therefore, the topic is of relevance in the current context of sea-ice and climate research. The manuscript is well written but I would appreciate if the authors would clarify some points. I also have some additional suggestions.

Thematic comments

**1 In the end of the introduction, it is not clear to the reader what precisely is the scope of the study and what is new about it. It is clear that the authors will describe the evolution of the sea-ice volume budget, in a similar way to Holland et al. (2010), with the method of Keen et al. (2013). However, it is not clear if the scope of the manuscript is to introduce the method for further application (as is suggested in the conclusion) or to draw conclusions from the ice volume budget evolution to improve the understanding of changes in the Arctic climate system as a whole. I would appreciate if the authors make this point very clear in the beginning. It is difficult to follow the story of the manuscript otherwise. The authors write on page 5: "although here we are also able to include individual components [. . .] volume budget". I suggest including this information in the introduction as it is a strong statement about what makes this study special.**

We intend the scope of the manuscript to be both the introduction of the method, illustrated by the application to the HadGEM2-ES model, and the investigation of the impact of the forcing scenario on the budget changes.

The abstract (p9) and introduction (p10-12) have been updated to clarify this, and the statement from p5 of the original manuscript has been incorporated into the revised introduction.

**2 The reference period is chosen as the years 1960-79. I would like the authors to comment why they chose this period and not the period 1960-1989 (as usually used in studies for IPCC assessment reports) or the period 1950-79 (to have at least 30 years).**
I find this period rather short to be a reference period. I wonder if they have tried other reference periods? And if yes, do they yield different results?

We have also considered changes w.r.t the 30 year period 1960-89, and the results and conclusions are almost identical. All the relevant figures have now been updated to use the longer reference period, and the text has been updated as appropriate. Some of the numbers quoted in the text have changed very slightly

**3 P4 L22-24: I do not understand why Eq. 1 should result in an ice volume that has to be converted back to effective ice thickness. As far as I understood, Eq. 1 gives thicknesses directly. I would appreciate if the authors could clarify this.**

Apologies, this was incorrect and this paragraph has been modified (p13 L8-10).

**4 P4 L30: Can the authors explain the especially steep decline of the winter sea-ice cover from 2080 onwards in the RCP8.5 scenario with their results? I would guess it has to do with the increase in water temperature inhibiting the formation of a winter sea-ice cover (see Bathiany et al. 2016). I would find interesting to hear if the authors have another explanation. It would be worth mentioning in the manuscript as well. On the same note, I would suggest that the greater decline in basal ice growth (P9 L26) is linked to the greater decline in ice area in RCP8.5 stated earlier in the manuscript. Maybe these two could be linked to make a statement about the processes at work here.**

Yes, it is most likely that the steeper decline in winter ice cover towards the end of the 21st century in RCP8.5 is associated with the warming ocean surface: certainly the DJF ocean top level temperature increases more rapidly towards the end of the integration. This is now mentioned in the text (p13 L18-21 and p20 L24-27)

**5 P7 L4: The lateral melting is not explicitly modeled. Do the authors have an idea of how important this term is? I could imagine that it is an important term in summer, as a component of the sea-ice albedo feedback.**

In a 'present day' (year 2000) long equilibrium run of our latest (CMIP6) model HadGEM3 GC3.1, the lateral melting term is important in the mean budget of the Arctic sea ice during JJA, when it is at most about 14% of the ocean to ice heat flux. It may become more important in a warming climate, and while this is outside the scope of this manuscript it will be possible to investigate this using data from CMIP6 scenario runs once they are available.
HadGEM2-ES does include an adjustment [*] to the ocean to ice heat flux when the ice concentration drops below 0.05, to provide a crude representation of increased lateral melting of small ice floes in the marginal ice zone.
[*] The heat flux is scaled by 0.05/ice_area so that the grid box integral of the flux becomes independent of the ice area .

In the revised manuscript we have now mentioned the lateral melting in the discussion (p22 L21-24)

**6 P7 L19-22: Holland found large differences between CMIP3 models. I would like the authors to comment on the implications of their findings for these differences or differences between CMIP5 models. In any case, I suggest moving this paragraph to**

the discussion in the end of the manuscript.

The point we intended to make here was that given Holland et al found that CMIP3 models did not even agree on the relative role of melt and growth in the ice decline, we might also expect considerable inter-model differences when we can break these terms down further for the CMIP6 models. We now mention this is the discussion (p23 L10-12)

**7 P8 L29: The authors write that the extra top melting is enhanced by reductions in the surface albedo. Do they infer this directly from the model simulation? I wonder if maybe longwave radiation also has an influence on surface melting (see e.g. Notz and Stroeve, 2016), for example through clouds and water vapor? I would like the authors to comment on that.**

Yes the LW also has an impact on the extra melting. We have added some extra text to show the relative magnitude of the SW and LW changes, and the albedo changes (p18 L10-21)

**8 P9 L2: I am not convinced that the in-situ warming of the ocean is only a consequence of the ice cover retreat in your model. Could it not also partly be due to a higher advection of oceanic heat from lower latitudes, as stated for example in Burgard and Notz, 2017? I would like the authors to comment on that.**

Yes, the advection of oceanic heat from lower latitudes does also play a role, and as shown in Burgard and Notz it is the main driver for the long term warming of the Arctic Ocean. However a seasonal analysis of the Arctic Ocean budget for HadGEM2-ES shows that during the spring (MAM) and summer (JJA), when large increases in the basal melting are seen, atmospheric surface fluxes are the major driver of warming, especially for the upper ocean. We have added this to the updated manuscript (p18, L31-34)

**9 The conclusion from Fig. 9 and Fig. 10 is that the changes in components of the ice volume budget are independent of the forcing and dependent on the remaining sea-ice area. I agree that this relationship is very clear. However, can the authors be sure that it is not rather dependent on the temperature? Several studies showed that the sea-ice area depends linearly on the air temperature (e.g. Winton, 2011; Mahlstein and Knutti, 2012) and cumulated $CO_2$ emissions (Notz and Stroeve, 2016). It might be worth having a look at these relationships as well to get a larger picture and maybe a stronger conclusion.**

Yes, there is a linear relationship between anomalies in the ice area and the near-surface air temperature in HadGEM2-ES, which holds for all the forcing scenarios considered here. We did consider plotting Fig 9 and Fig 10 using temperature instead of ice area. The main reason that we decided not to was because we felt that the changes in the budget terms and the changes in ice area were more directly and closely linked (eg smaller basal growth term in the ice budget because the growth is occurring over a smaller area). We agree that it would be advantageous to widen the discussion to include these relationships as well and have mentioned this in the discussion (p22 L32 – p15 L5)

**10 The last paragraph of the conclusion is somewhat unclear and is not very strong. This is not an advantage for the manuscript. I would suggest discussing a little more what makes this study special and what are its implications for future research. It is still not clear enough for me.**

We have re-written this section (p23 L6-14) to make it clearer.

Writing comments

**11 The Section 2.2 about model integration is interesting but I think there are too many details. The effect of the different $CO_2$ pathways on the temperature is what is important for the study. This effect can be seen well in Fig. 1. I therefore suggest that the authors leave in the reference to Moss et al. (2010) but that they leave out the bullet point list and the sentence "Fig. 1A of Caesar et al. [. . .] scenarios."**

This text has now been removed (p12 L25 – p5 L1)

**12 I suggest writing down the exact limits for the study area in an appendix/supporting information. This might be useful for the comparison with future studies.**

As it is a single sentence, we have added this information to the caption for figure 2 (p31)

**13 Section 3.1, P5-6: The bullet point list makes the text well readable. To keep consistent, maybe the authors could add some numbers to the three first points. There, the results are described qualitatively in contrast to the three last points, where they are described quantitatively.**

We have now added some numbers to the first three points so that this section has a consistent format. (p14 L16, L23 and L28-29)

**14 The transition between Section 3 and Section 4 is quite abrupt. I would suggest working on a more logical transition.**

Yes, agreed, we have added some text to improve this (p16 L1-3)

**15 In section 4.1., Fig. 5B is cited instead of Fig. 5A and vice versa. I suggest reading through this section carefully again.**

Thank you, and apologies for this. This has now been corrected (p16 L13-28)

**16 In section 4.2., the reader is pointed to several different figures while the rest of the manuscript is very structured (one paragraph = one figure description). In this case, it is helpful for the message to look at the different figures. However, I find difficult to follow the story from P9 L1 to P9 L22. I suggest to try reformulating the message in a clearer way.**

This section has been re-structured, so that the three budget components shown in figure 7c are discussed separately (p17 L22 – p20 L15). We have also added an extra pane to figure 7 to show the anomalies in the ice area (p41)

**17 P10 L26-27: The processes changing at the ice surface are listed and then "basal melting" is mentioned. Why?**

This was intended to refer to processes acting over the remaining ice, and the sentence how now been corrected (p21 L32 – p22 L1)

Technical comments

P1 L9-13: These two sentences are long and contain too much information. Reformulating might clarify the message.

This has been rewritten using shorter sentences. (p9, L12-14)

P3 L3: I suggest removing "for use in IPCC AR5". I think readers know the aim of CMIP5.

Yes, done at p11 L19.

P3 L9: West et al., 2017 is cited. In the references, it is marked as "in prep.". I think they can therefore not be cited it in this context then.
This work is now under review in The Cryosphere Discuss: the reference has been updated accordingly (p27 L18-20)

P3 L12: Replace "as that used" by "as the one used"
Done  (p11 L28)

P4 L6: Remove comma after the Moss et al., 2010 reference
P4 L15-16: The sentence is long. I suggest cutting after "scenario)" and starting the next sentence with "Fig.1".
Done (p13 L2)

P6 L25: The sentence is too long. I suggest stopping after "loss" and starting the next sentence with "The ice decline arises"
Done (p15 L19-20)

P6 L27: Add "seen in Fig. 3b" after "thickness".
Done (p15 L21)

P6 L29: The sentence is long. I suggest stopping after "line." and starting next sentence with "During"
Done (p7 L23)

P7 L8: Replace "and also how the seasonal cycle changes" by "and the changes in seasonal cycle".
Done (p16 L7-8)
P7 L27: "s" missing after 2040
P8 L 24-26: This sentence is too long. I suggest reformulating it to clarify the message.
Section 4.2 has been re-written, and this sentence is no longer there.

P8 L31-32: Can the authors reformulate this sentence? I do not understand it.
Section 4.2 has been re-written, and this sentence is no longer there.

P9 L9: add "process" between "this" and "that"
Section 4.2 has been re-written, and this sentence is no longer there.

P11 L2: I suggest changing "and reduced basal growth during autumn/early winter" to "and in autumn/early winter due to reduced basal growth".
Done (p22, L9)

P11 L15-18: This sentence is too long and unclear. I suggest reformulating it to clarify the message.
Much of section 5 has been re-written, and this sentence is no longer there.

Figures
Fig. 1: I suggest marking or shading the reference period In the caption, replace "HadGEM2ES" by "HadGEM2-ES"

Done (p29)

Fig. 6: Have the authors looked into the period 2080-2099? Are the changes still
similar? If not, would they bring additional information for the study?
5    Having looked again at this later period we have now added an extra pane to figure 6 to show the seasonal cycle for
2080-89 (p39), and this is referred to in sections 4.1 and 4.2.

Fig. 7: Add in the caption for (b) that this is 2010-2019.
Done (p41)

Fig. 9: 1960-79 instead of 1960-9
Updated to 1960-89 (p45)

References
15    Arctic is written "arctic" in most of your references. I suggest reading through them
carefully again.
The references have been checked and updated.

We thank the reviewer for reading our manuscript, and for his/her comments. Our updated responses are included below in blue text. The page numbers refer to the 'tracked-changes' section of this document, showing the alterations from the original version of the manuscript.

General Comments

In this work the authors decompose in the coupled climate model HadGEM2-ES the global Arctic sea ice volume budget over the late 20th and 21st century into its main components – top melt, basal growth, basal melt, frazil ice formation, advection, snowfall less sublimation.

In many ways this study appears as a follow up study of the earlier Keen et al., 2013 paper - see section 5 on 'modelled heat budget of the Arctic snow and ice' but instead of taking a local (per unit ice area) analysis here the authors present a global perspective that presents the advantage of explaining the mechanisms that control sea ice volume decline at the Arctic basin scale.

The main results of this study are:

- To present a detailed methodology of how to analyse the HadGEM2-ES Arctic sea ice volume budget components at the basin scale - To characterise and rank in order of importance the different terms controlling the seasonal and inter-annual sea ice growth (and melt) - To show that the changes in the volume budget are a function of the sea ice cover and not of the speed at which the sea ice retreats

My overall impression is that there is nothing fundamentally wrong with this paper but that at the same time that it does not contribute to any significant advances in the field.

We feel that the type of study outlined here is now necessary in order to understand inter-model differences in projected future ice decline: we need to consider changes in the underlying processes as well as looking at how the ice state changes. We also feel that there are a number of novel aspects to our study, some of which are summarised below. However we recognise from this review, and from comments from reviewer 1, that we were clear enough about the scope and interest of the study in the original version of the manuscript.

To remedy this we have updated the Abstract, Introduction and Summary and Discussion to better clarify the scope of the study, and the novel aspects.

I encourage the authors to explore one of the following possible extensions of their work in order to give it a wider audience:

- Explore impact of sea ice physics even at a simple level. Comparing results with results from HadGEM1 analyzed in Keen et al., 2013 could be informative. While it would be difficult to separate the impact of the different physics in the two models on the total volume budget it would show how model developments modify our understanding of the drivers of sea ice decline

We have applied the same analysis to HadGEM1, but the sea ice physics in these two models is essentially the same, so this comparison would not show the impact of different sea ice physics. The two models showed very similar budget change, with the main differences being due to different rates of ice decline. While we *are* interested in the impact of the ice decline in the changing budget, we felt this was better illustrated using one model and a range of different forcing scenarios.

. - Compare the model results with other climate models.
In that sense the reader would get a better sense of inter-model variability. The authors
suggest that their methodology is appropriate to analyse other models. Why not do it?

This will be possible using the diagnostics from CMIP6 models, and we plan to do this once the data is available.

- If these options appear too ambitious the authors may at least consider improving the
quality of the figures and explain in greater details how the decompositions presented
in those figures help explain the future evolution of the sea ice cover and its role in the
climate as a whole. For example what can we learn about the changing climate based
on seasonal changes in the different terms in the volume budget. Similarly what do the
figures 9 and 10 on the changes of effective thickness as a function of sea ice area
tells us about climate change in the Arctic and beyond.

We have expanded the discussion in section 4.2 (p17-20) to include more about the processes in the wider climate and their impact on the declining sea ice. In the Summary and Discussion we now link the changes shown in figures 9 and 10 with global changes in near-surface temperature and cumulative $CO_2$ emissions (p22 L31 – p23 L5)

**Summary of changes made to manuscript**

- We have updated the abstract and introduction to clarify the scope and novel features of this study.

- We have re-structured section 4.2 so that is it clearer, and have provided more information about processes in the wider Arctic that contribute to the changes in the volume budget components.

- We have updated all the figures to use a longer reference period, and added an extra pane to each of figures 6 and 7.

- We have updated the Summary and discussion to clarify the novel features of our work, and the implications for future research.

[revised manuscript text omitted]

15 atmosphere, but predominantly due to changes in the impact of cloud (71%). The outgoing SW decreases partly because there is less incoming SW, but predominantly because of the reduced surface albedo (67%). In common with other CMIP5 models, incoming LW increases due to the higher levels of $CO_2$ in the atmosphere (Notz and Stroeve, 2016) found a robust linear relationship between incoming non-shortwave fluxes and cumulative $CO_2$ emissions for CMIP5 models). Cloud modifies the amount of LW reaching the surface, but there is little change in the overall impact of cloud on downward LW as

[revised manuscript text omitted]

**5 Title of the manuscript**

Angie A. Aman[1], B. Bradley Bman[1], Carl Cman[2]

[1]Department example, University example, city, postal code, country
[2]Laboratory example, city, postal code, country

*Correspondence to*: Angie A. Aman (aman@email.com)

**Abstract.** Please use only the styles of this template (MS title, Authors, Affiliations, Correspondence, Normal for your text, and Headings 1–3). Figure 1 uses the style Caption and Fig. 1 is placed at the end of the manuscript. The same is applied to tables (Aman et al., 2014; Aman and Bman, 2015) adipiscing elit. Mauris dictum, nibh ut condimentum pharetra, quam ligula varius est, sed vehicula massa erat ut metus. In eget metus lorem. Fusce vitae ante dictum, elementum sem non, lacinia dui. Integer tellus tortor, convallis et aliquam non, dictum vel mauris. Quisque maximus mollis dui, a mollis mauris vehicula in. Duis dui ligula, suscipit ac lectus vitae, fringilla euismod diam.

**1 Section (as Heading 1)**

Suspendisse a elit ut leo pharetra cursus sed quis diam. Nullam dapibus, ante vitae congue egestas, sem ex semper orci, vel sodales sapien nibh sed lectus. Etiam vehicula lectus quis orci ultricies dapibus. In sit amet lorem egestas, pretium sem sed, tempus lorem.

**1.1 Subsection (as Heading 2)**

Quisque cursus massa sed urna congue, ac convallis neque consectetur. Proin faucibus neque non metus mollis, suscipit pretium nisl blandit. In hac habitasse platea dictumst. Nam laoreet augue eu odio eleifend, non posuere quam pulvinar. Integer sit amet leo vitae nisl facilisis tristique.

**1.2 Subsection (as Heading 2)**

Ut rutrum, sapien et vulputate molestie, augue velit consectetur lectus, bibendum porta justo odio lobortis ligula. In in urna nec arcu iaculis accumsan nec et quam. Integer ut orci mollis, varius justo vitae, pellentesque leo. Vestibulum eu finibus nisl. Cras ac arcu urna. Duis ut pellentesque urna.

**1.2.1 Subsubsection (as Heading 3)**

In placerat dictum urna ut interdum. Etiam vel nibh vulputate, scelerisque purus in, congue eros. Pellentesque at nisi at nunc sagittis cursus. Mauris euismod tellus at mi tempor, sit amet finibus ante tincidunt. Aenean id ornare neque. Cras ut sapien quis erat pretium ultricies. Integer vulputate ante nec elementum tristique. Ut. Lorem ipsum dolor sit amet, consectetur adipiscing elit. Mauris dictum, nibh ut condimentum pharetra, quam ligula varius est, sed vehicula massa erat ut metus. In eget metus lorem. Fusce vitae ante dictum, elementum sem non, lacinia dui. Integer tellus tortor, convallis et aliquam non, dictum vel mauris. Quisque maximus mollis dui, a mollis mauris vehicula in. Duis dui ligula, suscipit ac lectus vitae,

fringilla euismod diam. Suspendisse a elit ut leo pharetra cursus sed quis diam. Nullam dapibus, ante vitae congue egestas, sem ex semper orci, vel sodales sapien nibh sed lectus. Etiam vehicula lectus quis orci ultricies dapibus. In sit amet lorem egestas, pretium sem sed, tempus lorem. Quisque cursus massa sed urna congue, ac convallis neque consectetur. Proin faucibus neque non metus mollis, suscipit pretium nisl blandit. In hac habitasse platea dictumst. Nam laoreet augue eu odio

5   eleifend, non posuere quam pulvinar. Integer sit amet leo vitae nisl facilisis tristique calculated following Eq. (1):

$$Y = \frac{\Delta M_0}{\Delta[\text{isoprene}]} \, , \tag{1}$$

where $\Delta M_0$ is Ut rutrum, sapien et vulputate molestie, augue velit consectetur lectus, bibendum porta justo odio lobortis ligula. In in urna nec arcu iaculis accumsan nec et quam. Integer ut orci mollis, varius justo vitae, pellentesque leo. Vestibulum eu finibus nisl. Cras ac arcu urna. Duis ut pellentesque urna. In placerat dictum urna ut interdum. Etiam vel nibh

10   vulputate, scelerisque purus in, congue eros. Pellentesque at nisi at nunc sagittis cursus. Mauris euismod tellus at mi tempor, sit amet finibus ante tincidunt. Aenean id ornare neque. Cras ut sapien quis erat pretium ultricies. Integer vulputate ante nec elementum tristique. Ut.

[Figure]

**Figure 2: The logo of Copernicus Publications.**

---

## Referee Report (RR1)

Second review for Keen and Blockley
**Investigating future changes in the volume budget of the Arctic sea ice in a coupled climate model**
Submitted to *The Cryosphere*

I thank the authors for answering my questions clearly and taking into account my comments. The manuscript has a clearer purpose and an improved structure compared to the last version. I also appreciate that the results are underlined more quantitatively than before in the text.
Besides the comments below, the science behind the manuscript is clear enough. However, I suggest that the authors use more conjunctions to improve the readability of this very technical and descriptive study.
Still, overall, the manuscript is nearly ready for publication and I only recommend minor revisions.

**Thematic comments**

**#1** P9L12: I would agree that increased CO2 has an influence on incoming longwave radiation, but I do not know how it has an impact on incoming shortwave. Could the authors clarify?

**#2** P9L16 : I suggest replacing "non-shortwave" by longwave. Notz and Stroeve (2016) show that the highest effect comes from the incoming longwave radiation.

**#3** P9L16-19: The reasoning of this part is not clear: There is not much change in downward longwave, there is an increase in the outgoing longwave, but the net change is an increase in net downward longwave. There is one explanation missing to make this couple of sentences logically linked.

**#4** P11 L5-10: Is there a reason why RCP4.5 is not really mentioned?

**Style**

**#5** In contrast to a flowing story, the manuscript reads mostly as a succession of figure/process descriptions. I suggest to work on clear transitions and on highlighting logical links and contrasts.

**#6** Several times, the HadGEM2-ES is described as a CMIP5/AR5 model. I suggest stating it just once in the beginning.

**#7** Especially in the conclusion/discussion, "for HadGEM2-ES" is repeated many times. I think it is clear that results stated there are for HadGEM2-ES unless specified otherwise. I suggest going through this part again and remove some of these mentions.

**Technical comments**

P8 L1 : it begin**s** (s is missing)

P9 L30: I suggest starting the sentence with "however" to make clear that there is a contrast here.

Figure 6: I suggest writing the corresponding represented period as a title for each subplot.

---

## Author Response (AR2)

**Response to anonymous reviewers**

We thank both reviewers for reading our revised manuscript, and for their comments. Our responses are included below in blue text. The page numbers refer to this document, showing the alterations from the revised version of our manuscript.

**Reviewer #1**

I thank the authors for answering my questions clearly and taking into account my comments. The manuscript has a clearer purpose and an improved structure compared to the last version. I also appreciate that the results are underlined more quantitatively than before in the text.
Besides the comments below, the science behind the manuscript is clear enough. However, I suggest that the authors use more conjunctions to improve the readability of this very technical and descriptive study.
Still, overall, the manuscript is nearly ready for publication and I only recommend minor revisions.

**Thematic comments**

**1 P9L12: I would agree that increased CO2 has an influence on incoming longwave radiation, but I do not know how it has an impact on incoming shortwave. Could the authors clarify?**
We meant the water vapour changes due to the increased CO2, but this was unclear and the sentence has been re-worded. (P12L18-19)

**2 P9L16 : I suggest replacing "non-shortwave" by longwave. Notz and Stroeve (2016) show that the highest effect comes from the incoming longwave radiation.**
Yes, this has been changed (P12L24)

**3 P9L16-19: The reasoning of this part is not clear: There is not much change in downward longwave, there is an increase in the outgoing longwave, but the net change is an increase in net downward longwave. There is one explanation missing to make this couple of sentences logically linked.**
We have re-worded this and hopefully it is now clearer (P12L24-28)

**4 P11 L5-10: Is there a reason why RCP4.5 is not really mentioned?**
This was an omission – RCP4.5 is now mentioned here (P4L22)

**Style**

**5 In contrast to a flowing story, the manuscript reads mostly as a succession of figure/process descriptions. I suggest to work on clear transitions and on highlighting logical links and contrasts.**
We have added some text to emphasise links and contrasts, and hope that this has made the text flow better.

**6 Several times, the HadGEM2-ES is described as a CMIP5/AR5 model. I suggest stating it just once in the beginning.**
We now only mention this in the abstract and model description, and in the introduction where we feel it is necessary to make it clear that HadGEM2-ES is not a CMIP6 model. We wanted to keep this in the model description so that it is clear for a reader who jumps straight to the model description.

**7 Especially in the conclusion/discussion, "for HadGEM2-ES" is repeated many times. I think it is clear that results stated there are for HadGEM2-ES unless specified otherwise. I suggest going**

through this part again and remove some of these mentions.
We have now reduced the number of specific mentions of HadGEM2-ES.

Technical comments
P8 L1 : it begins (s is missing) This has been corrected (P11L7)
P9 L30: I suggest starting the sentence with "however" to make clear that there is a contrast here.
This has been done. (P13L7)
Figure 6: I suggest writing the corresponding represented period as a title for each subplot.
This has been done. (P31)

Reviewer #2

Following previous reviews, the paper has been clarified and is acceptable for publication providing the minor comments below are addressed. Note that page and line numbers correspond to the edited document.

P10L29 cite papers, literature and ongoing community efforts on formalising CMIP6 SIMIP data requests.
We have added the Notz et al (2016) ref here, which we feel is the best source of information about the original data request and the background to its development, and also a link to access the most up to date version of the data request. (P5L26).

P12L20 You must state how many ensemble members are produced for each historical simulations. Do you analyse the same ensemble that will be released as part of CMIP6.
We already mention that there is an ensemble of 4 historical simulations (P7L12), and we have now clarified that there is also an ensemble of 4 for each forcing scenario (P7L15).
The data used here will not be released as part of CMIP6, as HadGEM2-ES was a CMIP5 model and these runs use the AR5 forcing scenarios. Some of the HadGEM2-ES model data is available through the CMIP5 data archive. For CMIP6, will be submitting data from our new model HadGEM3, forced with the AR6 scenarios.

Fig1. Show ensemble spread in shaded colours or with thin lines.
The individual ensemble members have been added as thin lines, and the caption updated accordingly (P23).

P15L20 budget sum not visible on Fig 4a. Could you show it multiplied by a factor 10 in dashed black on fig 4a or scaled on Fig3b for clarity?
We have added the budget sum x 10 to Figure 4a, and updated the text accordingly. (P9L29-P10L1 and P28)

Fig7c vertical legend should be area normalised change in effective ice
This has now been corrected. (P33)

P18L33 "...a budget analysis of the upper ocean shows..." This is mentioned as an aside but is an important point and deserves a little more explanations and/or a reference to other studies. For example is this pattern homogeneous throughout the Arctic or regionally varying (i.e. competition of atmospheric vs oceanic forcing in the Barents Sea).
We have added a reference to Graham and Vellinga (2012), whose approach we followed for this ocean budget analysis, and have added a little more text (P13L9-12). This is an interesting topic – Graham and Vellinga do mention that the loss during the autumn and winter of the heat gained during the summer may not be homogeneous but we feel that further investigation of this is outside the scope of this study.

P20L8 Make it crystal clear when you talk about effective Arctic wide melt and growth and when you talk about normalised or local changes. I suggest you use the term local when discussing normalised quantities as in Keen 2013.
We have adopted this suggestion, and now refer to the normalised quantities as local quantities from section 4.2 onwards.

Also add paragraph summarising similarities and differences with this earlier work. Are the results fully consistent? If not why, model differences or otherwise?

The HadGEM2-ES and HadGEM1 volume budgets are very similar. A paragraph has now been added in Section 3.1 to mention this, and to summarise the differences (P10L4-9)

P20L12 I thought more ice is melted at the top (i.e. Fig 5a).

There is more local melting at the surface as the runs progress, but less total surface melting due to the reductions in ice area. We have modified the wording in this paragraph to hopefully make this clearer (P14L13)
In Figure 5 the top melting term is positive because its impact on the volume budget is a net gain in ice.

Fig9 Nice plots but it would make sense to add standard deviation for pre-industrial run also in the x-direction (i.e. for ice area). Make it clearer if those scatter plot include all ensemble runs or are based on the mean values. Would the plot look different if x-axis was change in ice area or if the y-axis was effective ice thickness (and not change in...)
Thank you for this suggestion – we have added the standard deviation in ice area to both Fig 9 and Fig 10. The plots include all the ensemble runs, and this has been clarified in the caption. (P36 and P38). The plots look very similar if the x-axis is changed to be "change in ice area", or if the y-axis becomes "effective ice thickness".

Finally here are some general comments or a wish list that could help raise the impact of the paper. Could the authors make their model runs available to the readers in line with TC policy.
We have added a statement about data availability (P17 L5-6)

It would also have been nice to have more information on regional seasonal patterns and trends as supplementary information.
While we agree that a regional analysis would be interesting, we feel it is outside the scope of this work.

The quality of the figures is not perfect (for example arguably it would be clearer to show information on model spread as shaded areas or continuous thin lines rather than individual scatter points).
We now show model spread by plotting individual ensemble members as continuous thin lines in Fig. 1 and Fig. 3 (P23 and P26).

Overall no comparison is made with Keen et al. 2013 study. I think this is needed to put this work in context and show how model developments over the past years affect (or not) climate predictions in the Arctic.
As mentioned above, we have now added a paragraph mentioning Keen et al, and that the two models have a very similar mean budget (P10L4-9). As HadGEM1 and HadGEM2-ES have near-identical sea ice physics, we anticipate that the impact of sea ice model developments on future Arctic climate predictions will be more readily investigated using the forthcoming CMIP6 models.

[revised manuscript text omitted]

---

## Author Response (AR3)

**"Investigating future changes in the volume budget of the Arctic sea ice in a coupled climate model" by Ann Keen and Ed Blockley**

**Response to Editor (in blue)**

Comments to the Author:
Dear Ann, dear Ed,

thanks again for submitting this work to The Cryosphere. I am happy to now accept it for publication.

In reading the final version, I noted however to instances where an additional clarification would be helpful. Please consider including one or two sentences for the following to items:

1. Given that most climate models output the "equivalent ice thickness" (Hibler, 1979), which is the grid-cell average thickness of the ice including the zero-ice thickness of any open water in that cell, I would find it helpful to clarify that your "effective ice thickness" is the real ice thickness not including any contribution from open water.

2. Frazil ice:It might be helpful to clarify how the frazil ice contributes to the thickness of the existing ice. This is because one would usually expect a higher ice growth per unit area of open water than per unit area of existing ice, so one would expect a higher ice growth of frazil than of basal ice. I assume that the total growth of frazil ice is indeed higher in the model, but is than converted into ice-thickness change by averaging the frazil-ice growth over the area of the ice-covered part of the grid cell. This should be explained.

Thanks again for submitting this work, which will nicely inform upcoming papers based on hopefully widespread CMIP6 SIMIP output!

All the best, and have a good summer,

Dirk

Response from Authors:

Dear Dirk,

Thank you for your suggestions, which we have addressed as follows:

1) We have altered the text to clarify that our effective ice thickness is the total volume of ice within the domain of interest, divided by the area of the domain (also including the impact of any overlying snow).

2) We have added some text to clarify how our model deals with the frazil ice formation.

We hope these changes do make things clearer.

Thank you for your work as Editor of our manuscript, and we look forward to the new opportunities offered by the CMIP6 SIMIP data.

Wishing you also a good summer,

Ann and Ed.